# POMK regulates dystroglycan function via LARGE1-mediated elongation of matriglycan

Ameya S Walimbe[1], Hidehiko Okuma[1], Soumya Joseph[1], Tiandi Yang[1], Takahiro Yonekawa[1], Jeffrey M Hord[1], David Venzke[1], Mary E Anderson[1], Silvia Torelli[2], Adnan Manzur[2], Megan Devereaux[1], Marco Cuellar[1], Sally Prouty[1], Saul Ocampo Landa[1], Liping Yu[3], Junyu Xiao[4], Jack E Dixon[5], Francesco Muntoni[2,6], Kevin P Campbell[1]*

[1]Howard Hughes Medical Institute, Senator Paul D. Wellstone Muscular Dystrophy Specialized Research Center, Department of Molecular Physiology and Biophysics and Department of Neurology, Roy J. and Lucille A. Carver College of Medicine, The University of Iowa, Iowa City, United States; [2]Dubowitz Neuromuscular Centre, UCL Great Ormond Street Institute of Child Health & Great Ormond Street Hospital, London, United Kingdom; [3]Medical Nuclear Magnetic Resonance Facility, University of Iowa Roy J. and Lucille A. Carver College of Medicine, Iowa City, United States; [4]The State Key Laboratory of Protein and Plant Gene Research, School of Life Sciences, Academy for Advanced Interdisciplinary Studies, Peking-Tsinghua Center for Life Sciences, Peking University, Beijing, China; [5]Department of Pharmacology, Department of Cellular and Molecular Medicine, Department of Chemistry and Biochemistry, University of California, San Diego, San Diego, United States; [6]National Institute for Health Research Great Ormond Street Hospital Biomedical Research Centre, UCL Great Ormond Street Institute of Child Health, London, United Kingdom

*For correspondence:
kevin-campbell@uiowa.edu

Competing interests: The authors declare that no competing interests exist.

**Abstract** Matriglycan [-GlcA-$\beta$1,3-Xyl-$\alpha$1,3-]$_n$ serves as a scaffold in many tissues for extracellular matrix proteins containing laminin-G domains including laminin, agrin, and perlecan. Like-acetyl-glucosaminyltransferase 1 (LARGE1) synthesizes and extends matriglycan on $\alpha$-dystroglycan ($\alpha$-DG) during skeletal muscle differentiation and regeneration; however, the mechanisms which regulate matriglycan elongation are unknown. Here, we show that Protein *O*-Mannose Kinase (POMK), which phosphorylates mannose of core M3 (GalNAc-$\beta$1,3-GlcNAc-$\beta$1,4-Man) preceding matriglycan synthesis, is required for LARGE1-mediated generation of full-length matriglycan on $\alpha$-DG ($\sim$150 kDa). In the absence of *Pomk* gene expression in mouse skeletal muscle, LARGE1 synthesizes a very short matriglycan resulting in a $\sim$ 90 kDa $\alpha$-DG which binds laminin but cannot prevent eccentric contraction-induced force loss or muscle pathology. Solution NMR spectroscopy studies demonstrate that LARGE1 directly interacts with core M3 and binds preferentially to the phosphorylated form. Collectively, our study demonstrates that phosphorylation of core M3 by POMK enables LARGE1 to elongate matriglycan on $\alpha$-DG, thereby preventing muscular dystrophy.

## Introduction

The extracellular matrix (ECM) is essential for development, regeneration and physiological function in many tissues, and abnormalities in ECM structure can lead to disease (*Rowe and Weiss, 2008*; *Hudson et al., 2003*). The heteropolysaccharide [-GlcA-$\beta$1,3-Xyl-$\alpha$1,3-]$_n$ (called matriglycan) is a

scaffold for ECM proteins containing laminin-G (LG) domains (e.g. laminin, agrin, and perlecan) (*Yoshida-Moriguchi and Campbell, 2015*; *Hohenester, 2019*; *Michele et al., 2002*; *Ohtsubo and Marth, 2006*) and has the remarkable capacity to be tuned during skeletal muscle development and regeneration (*Goddeeris et al., 2013*). Over 18 genes are involved in the synthesis of the post-translational modification terminating in matriglycan (*Figure 1*), and defects in this process cause dystroglycanopathies, i.e. congenital and limb-girdle muscular dystrophies that can be accompanied by brain and eye defects. Like-acetyl-glucosaminyltransferase 1 (LARGE1) synthesizes matriglycan on the cell-surface glycoprotein, α-dystroglycan (α-DG) (*Inamori et al., 2012*). Addition of matriglycan enables α-DG to serve as the predominant ECM receptor in skeletal muscle and brain (*Yoshida-Moriguchi and Campbell, 2015*; *Hohenester, 2019*; *Jae et al., 2013*; *Yoshida-Moriguchi et al., 2010*; *Yoshida-Moriguchi et al., 2013*). Crystal structure studies have shown that a single glucuronic acid-xylose disaccharide (GlcA-Xyl) repeat binds to laminin-α2 LG4 domain (*Briggs et al., 2016*; *Hohenester et al., 1999*), and there is a direct correlation between the number of GlcA-Xyl repeats on α-DG and its binding capacity for ECM ligands (*Goddeeris et al., 2013*; *Inamori et al., 2012*). During skeletal muscle differentiation, LARGE1 elongates matriglycan to its full length for normal skeletal muscle function (*Goddeeris et al., 2013*). However, little is known about the mechanisms that control matriglycan elongation.

Complete loss-of-function mutations in the dystroglycanopathy genes abrogate synthesis of the post-translational modification terminating in matriglycan. Such mutations preclude addition of matriglycan and, thereby, cause the most severe form of dystroglycanopathy, Walker-Warburg syndrome (WWS), which is lethal *in utero* or within a day or two of birth (*Yoshida-Moriguchi and Campbell, 2015*; *Hohenester, 2019*; *Michele et al., 2002*; *Ohtsubo and Marth, 2006*). Protein *O*-Mannose Kinase (POMK) is a glycosylation-specific kinase that phosphorylates mannose of the core M3 trisaccharide (GalNAc-β1,3-GlcNAc-β1,4-Man) during synthesis of the *O*-mannose-linked polysaccharide ending in matriglycan (*Yoshida-Moriguchi and Campbell, 2015*; *Hohenester, 2019*; *Jae et al., 2013*; *Yoshida-Moriguchi et al., 2013*; *Zhu et al., 2016*). Interestingly, unlike with other dystroglycanopathy genes there are patients with complete loss-of-function mutations in POMK who suffer from mild forms of dystroglycanopathy (*Di Costanzo et al., 2014*; *von Renesse et al., 2014*), suggesting some expression of matriglycan without POMK. Here, we have used a multidisciplinary approach to show that phosphorylation of core M3 by POMK is not necessary for the LARGE1-mediated synthesis of a short, non-extended form of matriglycan on α-DG (~90 kDa) with reduced

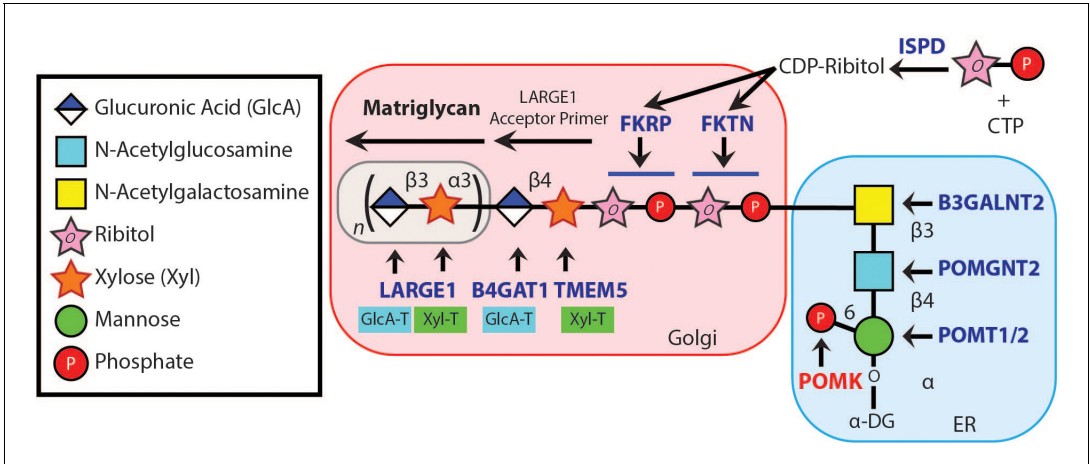

**Figure 1.** Synthesis of the α-DG Laminin-Binding Modification and Enzymes Involved. Synthesis of the laminin-binding modification begins with the addition of the core M3 trisaccharide (GalNAc-β3-GlcNAc-β4-Man) on α-DG by the sequential actions of Protein *O*-Mannosyltransferase 1 and 2 (POMT1/2), Protein *O*-linked Mannose *N*-Acetyl-glucosaminyltransferase 2 (POMGNT2), and β1,3-*N*-Acetylgalactosaminyltransferase 2 (B3GALNT2), in the ER. POMK phosphorylates the C6 hydroxyl of mannose after synthesis of core M3. The phosphorylated core M3 is further elongated in the Golgi by Fukutin (FKTN), Fukutin related protein (FKRP), Transmembrane Protein 5 (TMEM5), β1,4-Glucuronyltransferase 1 (B4GAT1), and Like-acetyl-glucosaminyltranserase 1 (LARGE1). Isoprenoid synthase domain-containing (ISPD) produces cytidine diphosphate (CDP)-ribitol in the cytosol, and this serves as a sugar donor for the reactions catalyzed by FKTN and FKRP. LARGE1 synthesizes matriglycan, which directly interacts with the LG domains of matrix ligands.

laminin-binding capacity; however, POMK activity is required for LARGE1 to generate full-length matriglycan on α-DG (~150 kDa). In the absence of the phosphorylated core M3, the non-extended matriglycan on ~90 kDa α-DG binds laminin and maintains specific force but cannot prevent eccentric contraction-induced force loss or skeletal muscle pathology. Furthermore, solution NMR studies demonstrated that LARGE1 directly interacts with core M3, binding preferentially to the phosphorylated form. Therefore, our study shows that phosphorylation of core M3 by POMK enables LARGE1 to elongate matriglycan on α-DG. Collectively, our work demonstrates a requirement for POMK in the LARGE1-mediated synthesis of full-length matriglycan and proper skeletal muscle function.

## Results

To determine if matriglycan can be expressed in the absence of POMK function, and therefore better understand the role of POMK in matriglycan synthesis, we studied skeletal muscle from a patient (NH13-284) with a homozygous POMK (D204N) mutation (*Figure 2A*) and congenital muscular dystrophy (CMD) accompanied by structural brain malformations. D204 serves as the catalytic base in the phosphorylation reaction catalyzed by the kinase (*Figure 2A*; *Figure 2—figure supplement 1*), and its mutation is predicted to eliminate POMK activity (*Figure 2—figure supplement 1*; *Zhu et al., 2016*). POMK activity from skin fibroblasts and skeletal muscle of patient NH13-284 (POMK D204N) was undetectable when compared to control fibroblasts and muscle, respectively (*Figure 2B*). Fibroblast LARGE1 activity and skeletal muscle B4GAT1 activity of patient NH13-284 were similar to those of a control (*Figure 2—figure supplement 2A and B*). Immunofluorescence analyses of POMK D204N muscle demonstrated partial immunoreactivity to IIH6 (anti-matriglycan), while the transmembrane subunit of DG, β-DG, was expressed normally in POMK D204N muscle (*Figure 2C*). Flow cytometry using IIH6 also demonstrated partial immunoreactivity in POMK D204N fibroblasts (*Figure 2—figure supplement 2C*). To test the effect of the POMK mutation on ligand binding, we performed a laminin overlay using laminin-111. Control human skeletal muscle showed the typical broad band of α-DG laminin binding centered at ~150 kDa range; in contrast, laminin binding at ~90 to 100 kDa range with reduced intensity was observed in POMK D204N skeletal muscle (*Figure 2D*).

To understand the biochemical basis of the ~90 to 100 kDa laminin binding in the absence of POMK activity, we targeted *Pomk* using LoxP sites and *Cre* driven by the *muscle creatine kinase* (*Mck*) promoter, or both the *Mck* promoter and the *paired box 7* (*Pax7*) promoter (*Figure 3—figure supplements 1* and *2*; *Brüning et al., 1998*; *Cohn et al., 2002*; *Han et al., 2009*; *Keller et al., 2004*) to generate muscle-specific *Pomk*-null mouse models. Histologic analyses of $Mck^{Cre}$; $Pax7^{Cre}$; $Pomk^{LoxP/LoxP}$ (M-POMK KO) quadriceps muscles revealed hallmarks of a mild muscular dystrophy (*Figure 3A*). Quadriceps muscle extracts of $Mck^{Cre}$; $Pomk^{LoxP/LoxP}$ mice showed reduced POMK activity compared to $Pomk^{LoxP/LoxP}$ muscle but had similar levels of LARGE1 activity (*Figure 3B and C*). M-POMK KO mice also showed reductions in 2-limb grip strength and body weight, and elevations in post-exercise creatine kinase (CK) levels compared to littermate control $Pomk^{LoxP/LoxP}$ mice (*Figure 3D*; *Figure 3—figure supplement 3*). Immunofluorescence analysis of M-POMK KO muscle showed that β-DG is expressed at the skeletal muscle sarcolemma (*Figure 3A*); however, like patient NH13-284 IIH6 immunoreactivity persisted in M-POMK KO muscle, but at a reduced intensity (*Figure 3A*).

We next examined *ex vivo* force production in extensor digitorum muscles (EDL) muscles of 18–20- week-old Control and M-POMK KO mice. EDL muscle mass and cross-sectional area (CSA) were reduced in M-POMK KO mice compared to control mice (*Figure 4A and B*). Additionally, M-POMK KO EDL absolute isometric tetanic force production was significantly lower than that of controls (*Figure 4C*). However, when normalized to muscle CSA, force production was comparable to control values (*Figure 4D*). We also sought to determine if M-POMK KO muscle could withstand repeated eccentric contractions. EDL muscles of M-POMK KO mice demonstrated greater force deficits after five and eight lengthening contractions (LC) and recovered to a lower level after 45 min compared to Control EDL (*Figure 4E*). Together, the isometric and eccentric contractile studies suggest that the M-POMK KO EDL muscles display a specific force similar to controls (*Figure 4D*); however, muscle integrity is compromised following the stress of repeated eccentric contractions, as displayed by the slow, but progressive decline in force production and hampered recovery (*Figure 4E*). Thus, the

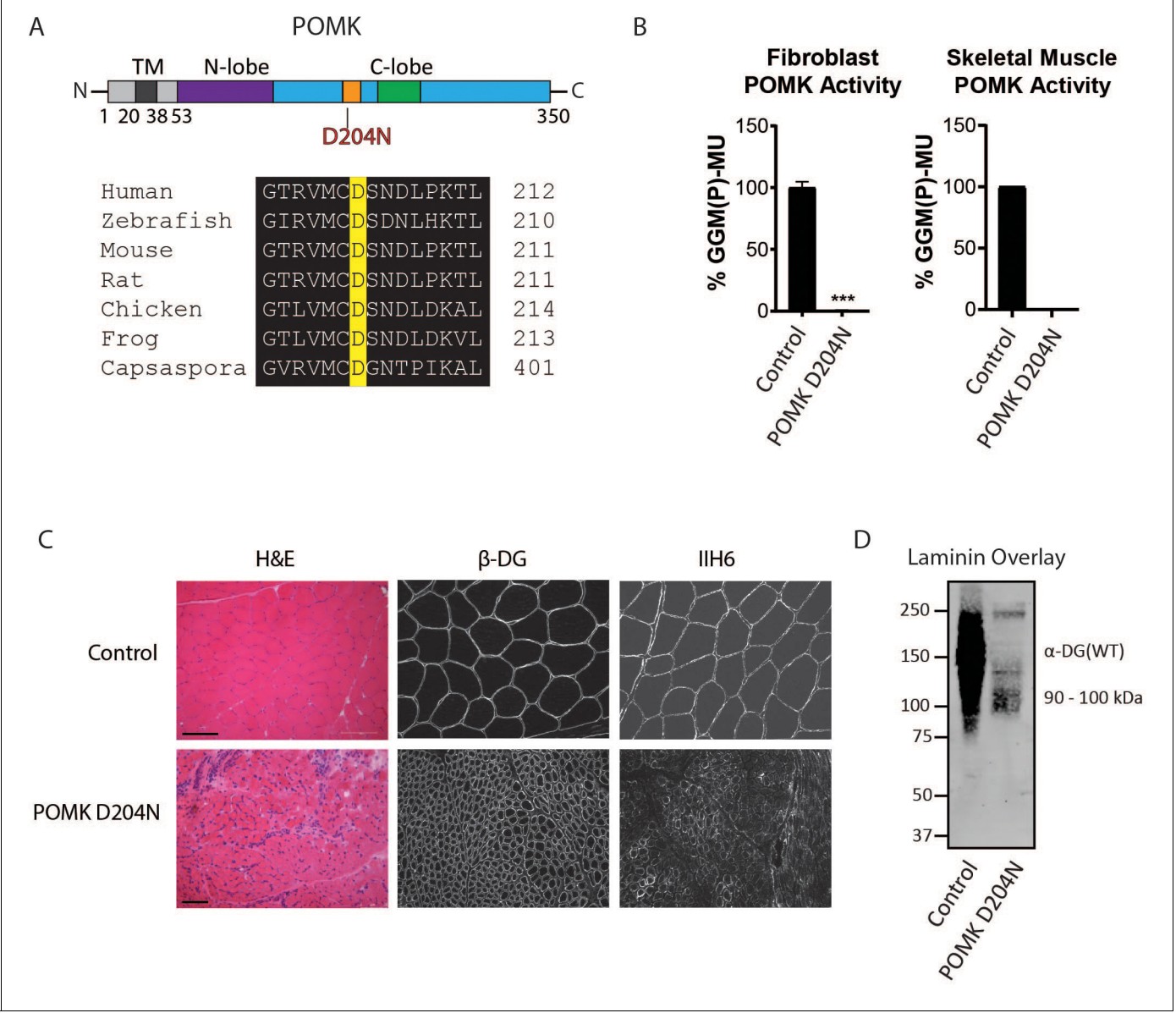

**Figure 2.** Characterization of a Patient with a Loss-of-Function Mutation in POMK. (**A**) (above) Human POMK consists of a transmembrane domain (TM) and a kinase domain (N-lobe and C-lobe). The kinase domain contains the catalytic loop (orange) and activation segment (green). (below) Alignment of protein sequences flanking the D204N mutation. The mutation alters a highly conserved aspartate that is the catalytic base of the phosphorylation reaction catalyzed by the kinase. (**B**) POMK activity in control and patient NH13-284 (POMK D204N) fibroblasts (left) and skeletal muscle (right). n = 3 experiments were performed in fibroblasts. Triple asterisks: statistical significance with Student's unpaired t-test (p-value<0.0001). Due to limited skeletal muscle, n = 1 experiment was performed. (**C**) Histology and immunofluorescence of control and POMK D204N skeletal muscle using IIH6 (anti-matriglycan) and a β-DG antibody. (Scale bars: Control- 200 μM, POMK D204N- 75 μM). (**D**) Laminin overlay of control and POMK D204N skeletal muscle.

The online version of this article includes the following figure supplement(s) for figure 2:

**Figure supplement 1.** Structural Modeling of POMK D204N Mutation.

**Figure supplement 2.** Supplemental Analysis of POMK D204N Fibroblasts and Muscle.

current results demonstrate that the short matriglycan in POMK-deficient skeletal muscle can maintain specific force but cannot prevent eccentric contraction-induced force loss or muscle pathology.

Biochemical analysis of control and M-POMK KO muscle showed a typical, lower molecular weight (MW) α-DG with anti-core DG antibody (*Figure 5A*), however, on laminin overlay, we

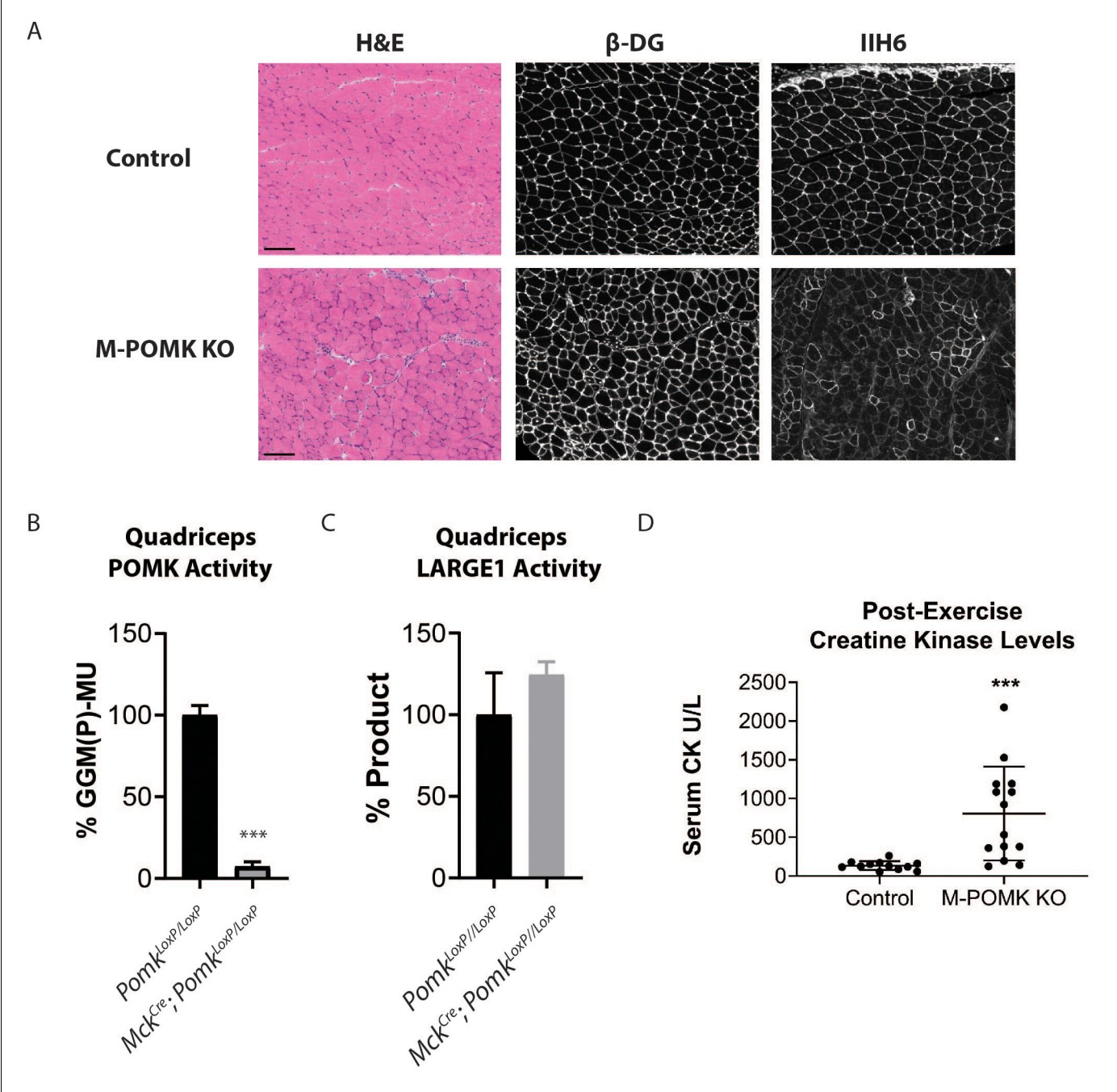

**Figure 3.** Mice with a Muscle-Specific Loss of *Pomk* Develop Hallmarks of a Mild Muscular Dystrophy. (**A**) H&E and immunofluorescence analyses using IIH6 (anti-matriglycan) and an anti-β-DG antibody of quadriceps muscles of 4–6 week-old *Pomk*$^{LoxP/LoxP}$ (Control) and *Mck*$^{Cre}$; *Pax7*$^{Cre}$; *Pomk*$^{LoxP/LoxP}$ (M-POMK KO) mice. Scale bars: 100 µM. (**B**) POMK and (**C**) LARGE1 activity in extracts of *Mck*$^{Cre}$; *Pomk*$^{LoxP/LoxP}$ and *Pomk*$^{LoxP/LoxP}$ quadriceps skeletal muscles. Triple asterisks indicate statistical significance using Student's unpaired t-test (p-value<0.0001, three replicates). (**D**) Creatine kinase levels of 8-week-old M-POMK KO and Control mice. p-values were calculated with Student's unpaired t-test. Triple asterisks: statistical significance with p-value<0.05 (p-value=0.0008), n = 12 Control and 14 M-POMK KO mice.

The online version of this article includes the following figure supplement(s) for figure 3:

**Figure supplement 1.** Schematic for Generation of Floxed Alleles of *Pomk*.

**Figure supplement 2.** Results of *Pomk*$^{LoxP/LoxP}$ Genotyping.

**Figure supplement 3.** Muscle-Specific *Pomk* Knockout Mice Have Reduced Grip Strength and Body Weight.

**Figure supplement 4.** Supplemental Biochemical Analysis of *Pomk*-null Skeletal Muscle.

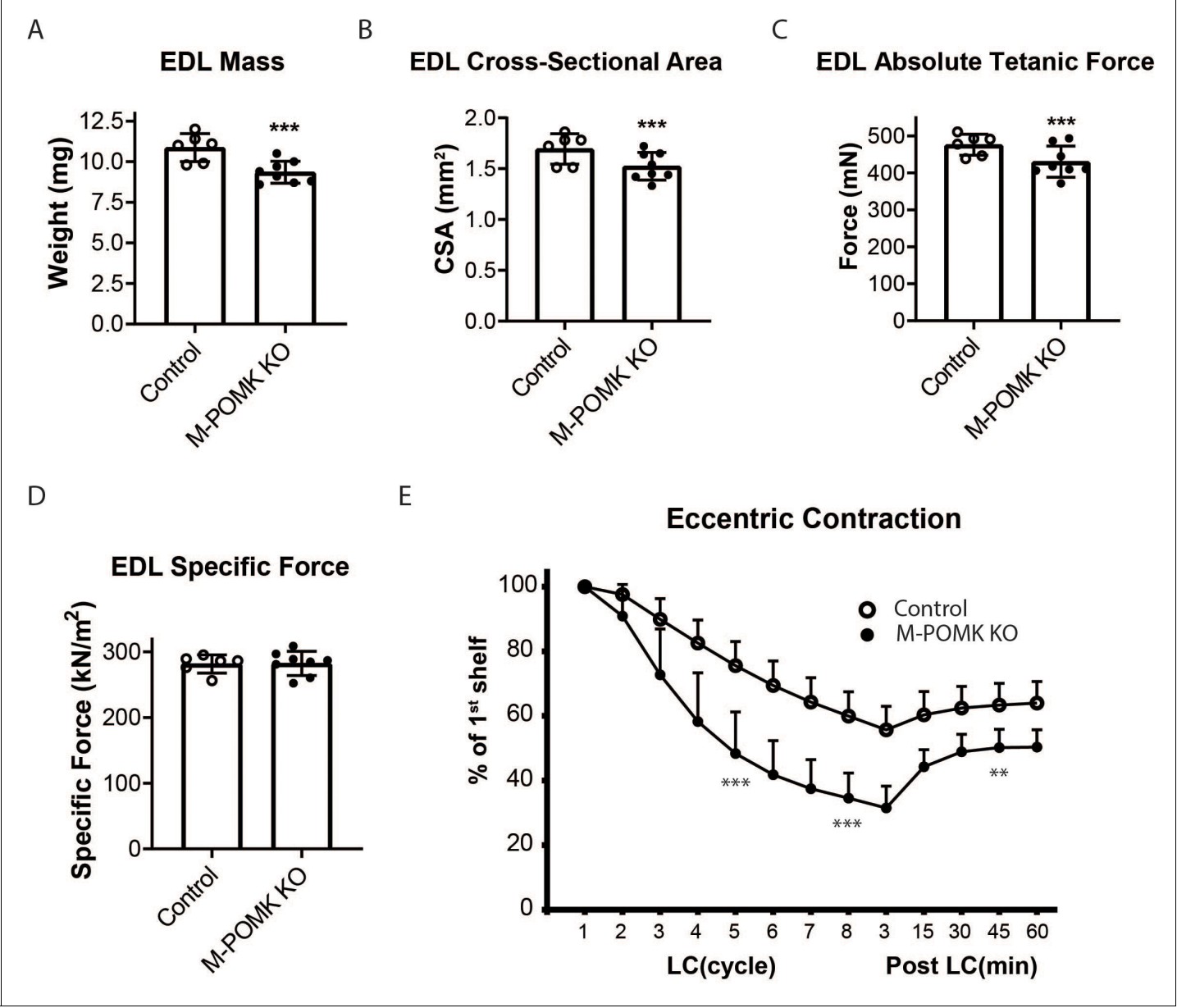

**Figure 4.** $Mck^{Cre}$; $Pax7^{Cre}$; $Pomk^{LoxP/LoxP}$ Extensor Digitorum Longus (EDL) Muscle Demonstrates Eccentric Contraction-Induced Force Loss. (**A**) Mass (milligrams) of $Pomk^{LoxP/LoxP}$ (Control) and $Mck^{Cre}$; $Pax7^{Cre}$; $Pomk^{LoxP/LoxP}$ (M-POMK KO) EDL muscles tested for force production. ***Statistical significance with Student's unpaired t-test with p-value<0.05 (p=0.0031). (**B**) Cross-sectional area (CSA) of EDL muscles. ***Statistical significance using Student's unpaired t-test with p-value<0.05 (p=0.0463). (**C**) Maximum Absolute Tetanic Force production by Control and M-POMK KO EDL muscles. ***Statistical significance using Student's unpaired t-test with a p-value<0.05 (p=0.0395). (**D**) Specific Force production in Control and M-POMK KO EDL muscles (p=0.921). (**E**) Force deficit and force recovery in Control (n=3) and M-POMK KO (n=4) mice after eccentric contractions. EDL muscles from 18- to 20-week-old male mice were tested and are represented by open (Control) or closed (M-POMK KO) circles. ***Statistical significance using Student's unpaired t-test (p-value<0.0001) compared to Control EDL at given LC cycle. **Statistical significance using Student's unpaired t-test (p-value=0.0027) compared to Control EDL at given LC cycle. Error bars represent SD.

observed laminin binding at 90–100 kDa (**Figure 5B**), similar to POMK D204N skeletal muscle (**Figure 2D**). IIH6 also showed binding at 90–100 kDa (**Figure 5C**). Solid-phase binding analyses of M-POMK KO and $Mck^{Cre}$; $Pomk^{LoxP/LoxP}$ skeletal muscle demonstrated a reduced binding capacity (relative $B_{max}$) for laminin-111 compared to control muscle (**Figure 5—figure supplement 1A**), but higher than that of $Large^{myd}$ muscle, which lacks matriglycan due to a deletion in $Large$.

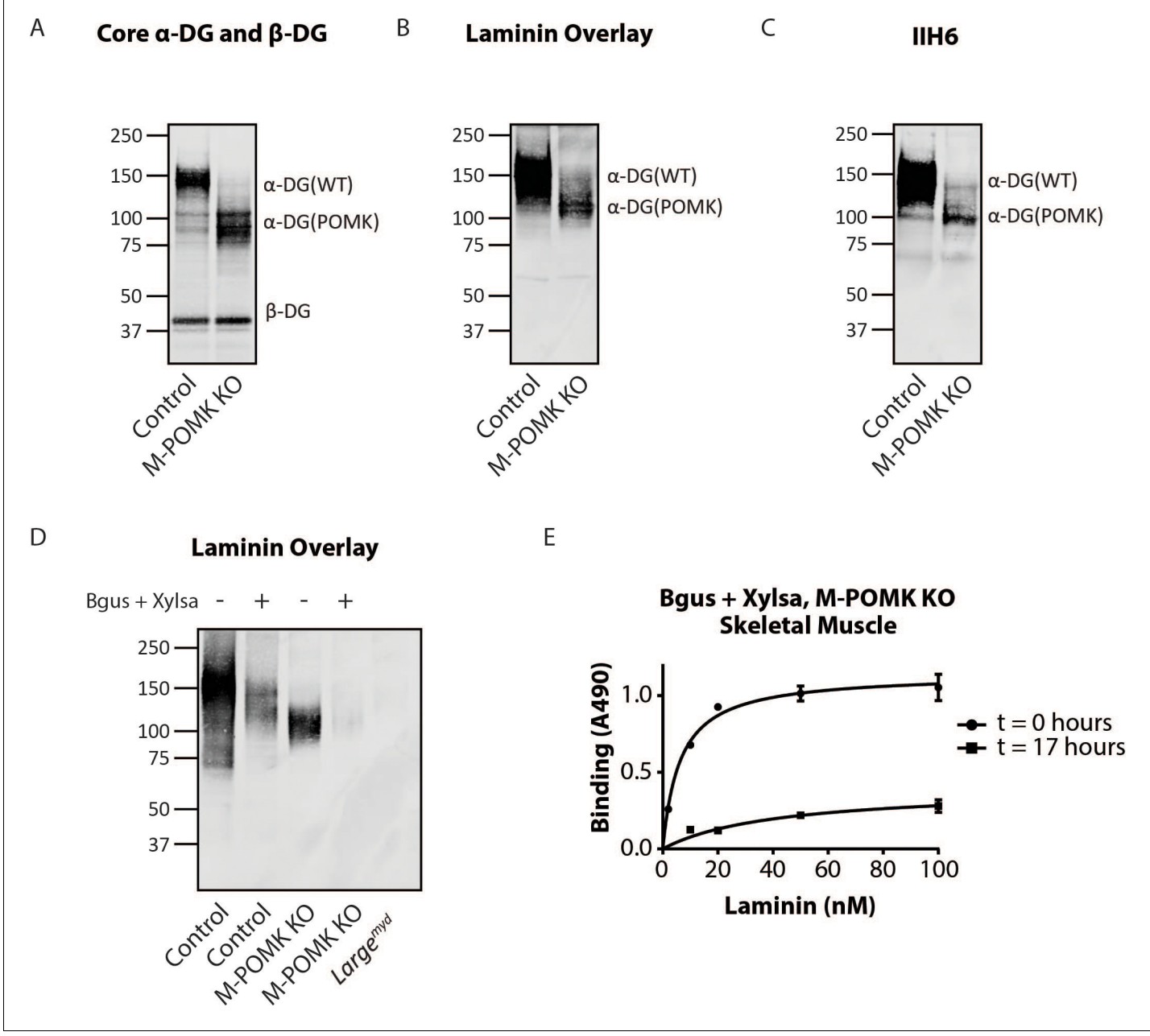

**Figure 5.** Mice with a Muscle-Specific Loss of *Pomk* Express Matriglycan. (**A**) Biochemical analysis of Control and M-POMK KO skeletal muscle. Glycoproteins were enriched from quadriceps skeletal muscles of mice using wheat-germ agglutinin (WGA)-agarose. Immunoblotting was performed with antibody AF6868, which recognizes core α-DG and β-DG (three replicates). (**B**) Laminin overlay of quadriceps muscles of Control and M-POMK KO mice (three replicates). (**C**) IIH6 immunoblotting of Control and M-POMK KO quadriceps muscle. (**D, E**) Laminin overlay (D) and solid-phase analysis (E) of skeletal muscles of M-POMK KO mice treated in combination with two exoglycosidases, α-xylosidase (Xylsa) and β-glucuronidase (Bgus) for 17 hr (three replicates).

The online version of this article includes the following figure supplement(s) for figure 5:

**Figure supplement 1.** Solid-Phase Binding Analyses of *Pomk*-null Skeletal Muscle.

**Figure supplement 2.** *Pomk*-null Muscle Expresses Matriglycan.

To determine if matriglycan is responsible for the laminin binding at 90–100 kDa in POMK-null muscle, we treated glycoproteins enriched from skeletal muscles of M-POMK KO and *Mck^{Cre}; Pomk^{LoxP/LoxP}* mice with two exoglycosidases, α-Xylosidase and β-Glucuronidase, which in combination digest matriglycan (*Figure 5—figure supplements 1B*, *2A and B*; *Briggs et al., 2016*). Laminin

overlay and solid-phase analysis showed a reduction in laminin binding from these muscles after dual exoglycosidase digestion (*Figure 5D and E*; *Figure 5—figure supplement 2A and B*).

To study the role of POMK further, we used human *POMK* KO HAP1 cells, which have undetectable levels of POMK activity and expression (*Figure 6A*; *Figure 6—figure supplement 1A*; *Zhu et al., 2016*). A mass spectrometry (MS)-based glycomic analysis of *O*-glycans carried by recombinantly-expressed DG mucin-like domain indicated the near complete absence of an MS peak at *m/z* 873.5 corresponding to phosphorylated core M3 *O*-glycan (*Figure 6D and E*; *Figure 6—figure supplement 2A and B*), consistent with an undetectable level of POMK activity in *POMK* KO HAP1 cells. Compared to WT HAP1 cells, immunoblots of *POMK* KO HAP1 cells showed a reduction in IIH6 immunoreactivity, a decrease in MW of core α-DG, and the presence of laminin binding at ~90 kDa on laminin overlay (*Figure 6C*; *Figure 6—figure supplement 1B and C*). Laminin binding on overlay was rescued only after adenoviral transduction with wild-type (WT) POMK (POMK WT), but not with POMK containing D204N (POMK D204N) or D204A (POMK D204A) mutations (*Figure 6C*). POMK D204N also lacked POMK activity *in vitro* but showed normal B4GAT1, B3GALNT2, and LARGE1 activity, thus confirming the pathogenicity of the D204N mutation (*Figure 6A and B*; *Figure 6—figure supplement 1D and E*).

To directly test if LARGE1 is required for synthesis of the 90 kDa laminin-binding glycoprotein in *POMK* KO HAP1 cells, we studied *POMK/LARGE1* KO HAP1 cells, which bear a CRISPR/Cas9-mediated deletion in *LARGE1* as well as *POMK*. *POMK/LARGE1* KO HAP1 cells demonstrated the absence of the laminin binding at 90 kDa (*Figure 7A*; *Figure 7—figure supplement 1A and B*), indicating that LARGE1 is required for the synthesis of the matriglycan responsible for laminin binding at 90 kDa. Moreover, *POMK/DAG1* KO HAP1 cells demonstrated a complete absence of laminin binding (*Figure 7A*) and IIH6 immunoreactivity at 90 kDa (*Figure 7—figure supplement 1C*), demonstrating that α-DG is the glycoprotein that binds laminin in the absence of POMK. We, therefore, refer to this glycoprotein as POMK-null α-DG (α-DG(POMK)). Since the length of matriglycan correlates with its binding capacity for ECM ligands (*Goddeeris et al., 2013*), we hypothesized that, given the MW of α-DG(POMK) at 90 kDa, the glycan must be shorter than full-length matriglycan, and therefore, have a lower $B_{max}$ for laminin. We measured the binding capacity of HAP1 α-DG using solid-phase binding assays. $B_{max}$ of α-DG(POMK) for laminin-111 was reduced compared to wild-type α-DG (α-DG(WT)) but was greater than that of α-DG from *LARGE1* KO HAP1 cells (*Figure 7B*). *POMK/DAG1* KO HAP1 cells showed a reduction in $B_{max}$ compared to *POMK* KO HAP1 cells, but similar to the low levels observed in *LARGE1* KO HAP1 cells (*Figure 7B*). These data indicate that a short, non-extended form of matriglycan is synthesized on α-DG(POMK), and this short form has a lower binding capacity for laminin-111, thus exhibiting a reduced level of α-DG receptor function.

After POMK phosphorylates core M3, Fukutin (FKTN) modifies GalNAc with ribitol-phosphate for synthesis of full-length matriglycan (*Figure 1*; *Yoshida-Moriguchi and Campbell, 2015*; *Hohenester, 2019*; *Kanagawa et al., 2016*). Overexpression in *POMK* KO HAP1 cells of ISPD, which synthesizes the substrate (CDP-ribitol) of FKTN (*Figure 1*), increases the amount of matriglycan (without changing its migration on SDS-PAGE) responsible for laminin binding at 90 kDa (*Figure 7—figure supplement 2A, B and C*; *Willer et al., 2012*; *Gerin et al., 2016*; *Riemersma et al., 2015*). HAP1 cells lacking both *POMK* and *ISPD* do not express matriglycan, and adenoviral transduction of these cells with ISPD restores the 90 kDa laminin binding (*Figure 7C*; *Figure 7—figure supplement 2D and E*). FKTN overexpression in *POMK* KO HAP1 cells also increased the 90 kDa laminin binding (*Figure 7—figure supplement 3A, B and C*). These experiments collectively support a requirement for CDP-ribitol for synthesis of the non-extended form of matriglycan. This synthesis also requires the N-terminal domain of α-DG (DGN) (*Hara et al., 2011a*; *Kanagawa et al., 2004*), as a DG mutant lacking the DGN (DGE) expressed in *POMK/DAG1* KO HAP1 cells did not show laminin binding at 90 kDa (*Figure 7—figure supplement 4A, B and C*). Similar experiments also indicated that synthesis of the non-extended matriglycan in HAP1 cells requires threonine-317 of the mucin-like domain of α-DG (*Figure 7—figure supplement 4A, B and C*).

Overexpression of LARGE1 can rescue the defect in matriglycan synthesis in distinct forms of CMD as well as in *LARGE1* KO HAP1 cells by generating very high molecular weight matriglycan (*Figure 7—figure supplement 5A*; *Barresi et al., 2004*). However, overexpression of LARGE1 in *POMK* or *POMK/LARGE1* KO HAP1 cells did not produce very high molecular weight matriglycan (*Figure 7D and E*; *Figure 7—figure supplement 5B, C and D*). Only the rescue of *POMK/LARGE1* KO HAP1 cells with POMK enabled LARGE1 to synthesize high molecular weight matriglycan

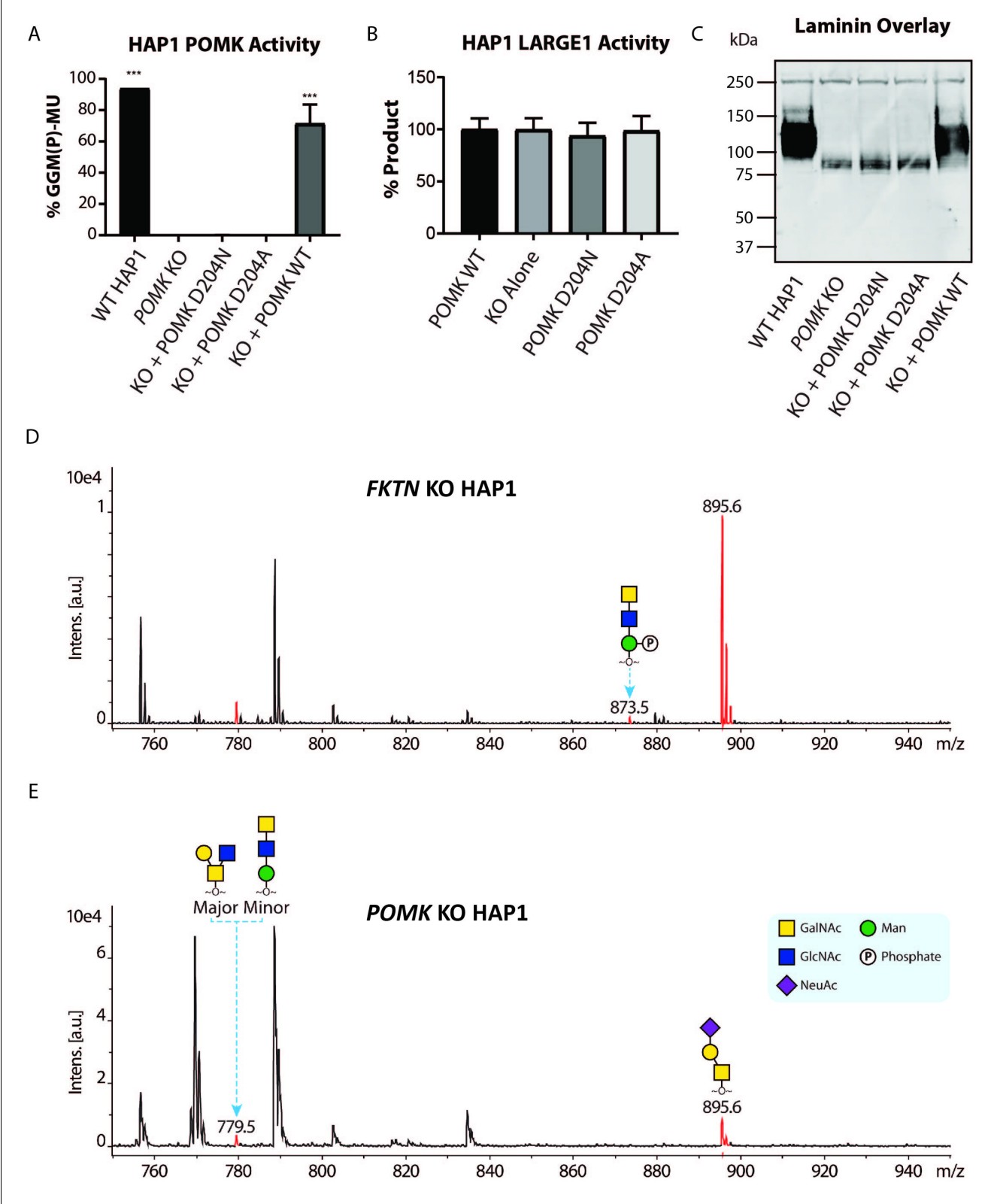

**Figure 6.** POMK D204N lacks Catalytic Activity. (**A**) POMK or (**B**) LARGE1 activity in *POMK* KO HAP1 cells transduced with adenoviruses encoding POMK D204N, POMK D204A, or POMK WT. Triple asterisks: statistical significance (p-value<0.0001) compared to *POMK* KO alone using one-way ANOVA with Dunnett's test for multiple comparisons (three replicates, 95% Confidence intervals for *POMK* KO vs. WT HAP1: −106.7 to −81.0, *POMK* KO vs. *POMK* KO + POMK WT: −84.25 to −58.54). (**C**) Laminin overlay of *POMK* KO HAP1 cells expressing the indicated POMK mutants. (**D, E**) Mass

*Figure 6 continued on next page*

*Figure 6 continued*

Spectrometry (MS)-based *O*-glycomic analyses of DG mucin-like domain (DG390TevHis) expressed in *Fukutin (FKTN)* (**D**) or *POMK* (**E**) KO HAP1 cells. *O*-glycans were released from the protein backbone and permethylated prior to matrix-assisted laser desorption/ionization time-of-flight (MALDI-TOF) analyses. MS peaks at *m/z* 779.5 (779.6) correspond to a mixture of core 2 and core M3 *O*-glycan, and at 873.5, phosphorylated core M3 *O*-glycan (red). MALDI-TOF is unable to determine anomeric or epimeric configurations of annotated *O*-glycans.

The online version of this article includes the following figure supplement(s) for figure 6:

**Figure supplement 1.** Supplemental Biochemical Analysis of POMK D204N and *POMK* KO HAP1 Cells.

**Figure supplement 2.** Mass spectra of *O*-glycans carried by a DG mucin-like domain model (DG390) expressed in *POMK* KO (**A**) or *Fukutin* (*FKTN*) KO (**B**) HAP1 cells.

(*Figure 7E*; *Figure 7—figure supplement 5D*). These findings indicate that LARGE1 requires phosphorylated core M3 to extend matriglycan on α-DG to its mature and high molecular weight forms.

To understand why phosphorylated core M3 is needed for LARGE1 to elongate matriglycan, we measured the binding affinity of LARGE1, as well as POMK, for the phosphorylated core M3 using solution NMR. We previously showed that the unphosphorylated core M3 binds to POMK with high affinity (*Zhu et al., 2016*). The mannose anomeric proton (Man H1) is well resolved and its intensity decreases only slightly with increasing POMK protein concentration (*Figure 8—figure supplement 1A*). By fitting the intensity changes of the Man H1 peak as a function of POMK concentration, we obtained a dissociation constant of >500 μM (*Figure 8C*; *Figure 8—figure supplement 1A and B*). These results indicate that, compared to the unphosphorylated core M3 of GGM-MU, the phosphorylated core M3 of GGMp-MU binds to POMK with a much weaker affinity. Then, we measured the binding affinities of LARGE1 for GGMp-MU and GGM-MU in a similar manner. Our results showed that LARGE1 binds with greater affinity to GGMp-MU compared to GGM-MU ($K_d = 11.5 \pm 1.2$ μM for GGMp-MU compared to $K_d$ >90 μM for GGM-MU) (*Figure 8A, B and D*). This indicates that the core M3 phosphate increases the binding affinity of LARGE1 for core M3 and could explain the ability of LARGE1 to elongate matriglycan in the presence of POMK.

## Discussion

POMK is a novel muscular dystrophy gene that phosphorylates mannose of the core M3 trisaccharide (GalNAc-β1,3-GlcNAc-β1,4-Man) on α-DG during synthesis of the *O*-mannose-linked polysaccharide ending in matriglycan. LARGE1 is responsible for the synthesis of matriglycan, and addition of matriglycan enables α-DG to serve as a predominant ECM receptor in many tissues, in particular, skeletal muscle and brain. Over eighteen genes are implicated in matriglycan synthesis, and complete loss-of-function mutations in these genes abrogate synthesis of the *O*-mannose linked modification and preclude the addition of matriglycan, thereby leading to dystroglycanopathies, congenital and limb-girdle muscular dystrophies with or without structural brain and eye abnormalities. Here, we have used a multidisciplinary approach to show that the absence of POMK activity does not preclude addition of matriglycan. Instead, in the absence of core M3 phosphorylation by POMK, LARGE1 synthesizes a short, non-extended form of matriglycan on α-DG (~90 kDa). However, in order to generate full-length mature matriglycan on α-DG (~150 kDa), LARGE1 requires phosphorylation of core M3 by POMK (*Figure 8—figure supplement 2A and B*).

Our study shows that the short form of matriglycan is able to bind to laminin with high affinity and thus enables α-DG(POMK) to function as an ECM receptor. Given the very small increase in apparent MW in α-DG(POMK) compared to α-DG from cells and muscle lacking LARGE1 (*Figure 5—figure supplement 2A*; *Figure 7—figure supplement 1A*; *Figure 8—figure supplement 3A*), the short, non-extended form of matriglycan likely contains few Xyl-GlcA repeats. However, it can still bind laminin since only a single Xyl-GlcA repeat is needed for laminin binding (*Briggs et al., 2016*), but it cannot function as an ECM scaffold. This short matriglycan likely attenuates muscular dystrophy in patient NH13-284 with a complete loss-of-function mutation in POMK, preventing the severe WWS phenotype that is observed in the complete absence of the other known dystroglycanopathy genes.

Muscle-specific POMK KO mice express the short, non-extended form of matriglycan on ~90 kDa α-DG and develop a mild muscular dystrophy phenotype. Muscle physiology studies demonstrate that the short matriglycan expressed in the absence of POMK can maintain specific force but cannot

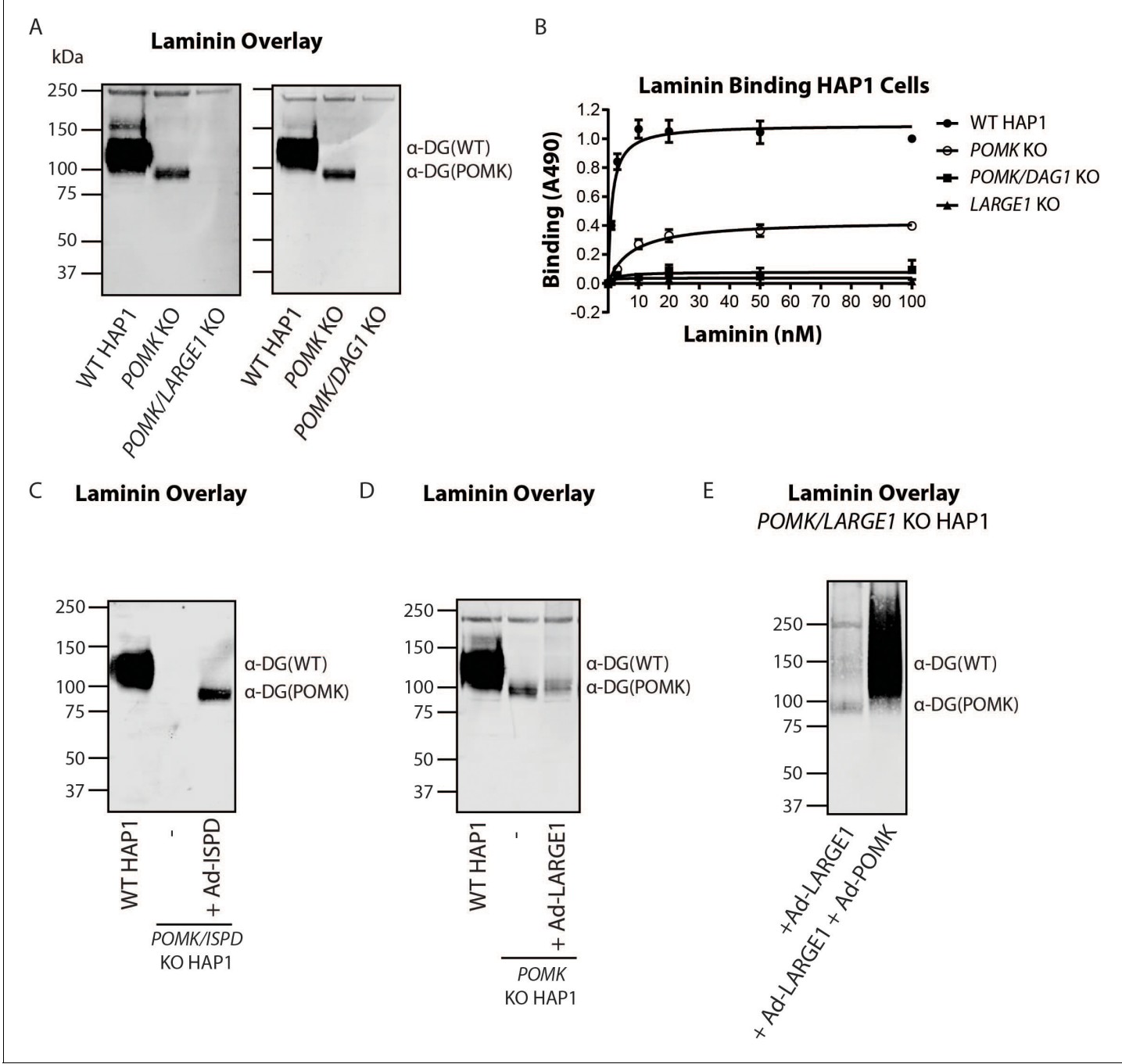

**Figure 7.** LARGE1 requires POMK to Elongate Matriglycan. (**A**) WT, *POMK* KO, and *POMK/LARGE1* KO HAP1 cells (left) or *POMK/DAG1* KO HAP1 cells (right) (three replicates). (**B**) Solid-phase analysis of WT, *POMK* KO, *POMK/DAG1* KO, and *LARGE1* KO HAP1 cells (three replicates). (**C, D, E**) Laminin overlays of the following KO HAP1 cells (three replicates): *POMK/ISPD* expressing Ad-ISPD (**C**); *POMK* expressing Ad-LARGE1 (**D**); *POMK/ LARGE1* expressing Ad-LARGE1 with or without Ad-POMK (**E**).

The online version of this article includes the following figure supplement(s) for figure 7:

**Figure supplement 1.** Supplemental Biochemical Analysis of *POMK/LARGE1* KO and *POMK/DAG1* KO HAP1 Cells.

**Figure supplement 2.** Requirement for Ribitol-Phosphate in the Synthesis of the Non-Extended Form of Matriglycan.

**Figure supplement 3.** Fukutin Overexpression Enhances Synthesis of the Non-Extended Matriglycan.

**Figure supplement 4.** T317 is Required for Synthesis of the Non-Extended Matriglycan.

**Figure supplement 5.** POMK Enables LARGE1-mediated Elongation of Matriglycan.

**Figure supplement 6.** Supplemental Characterization of POMK-null Matriglycan Synthesis.

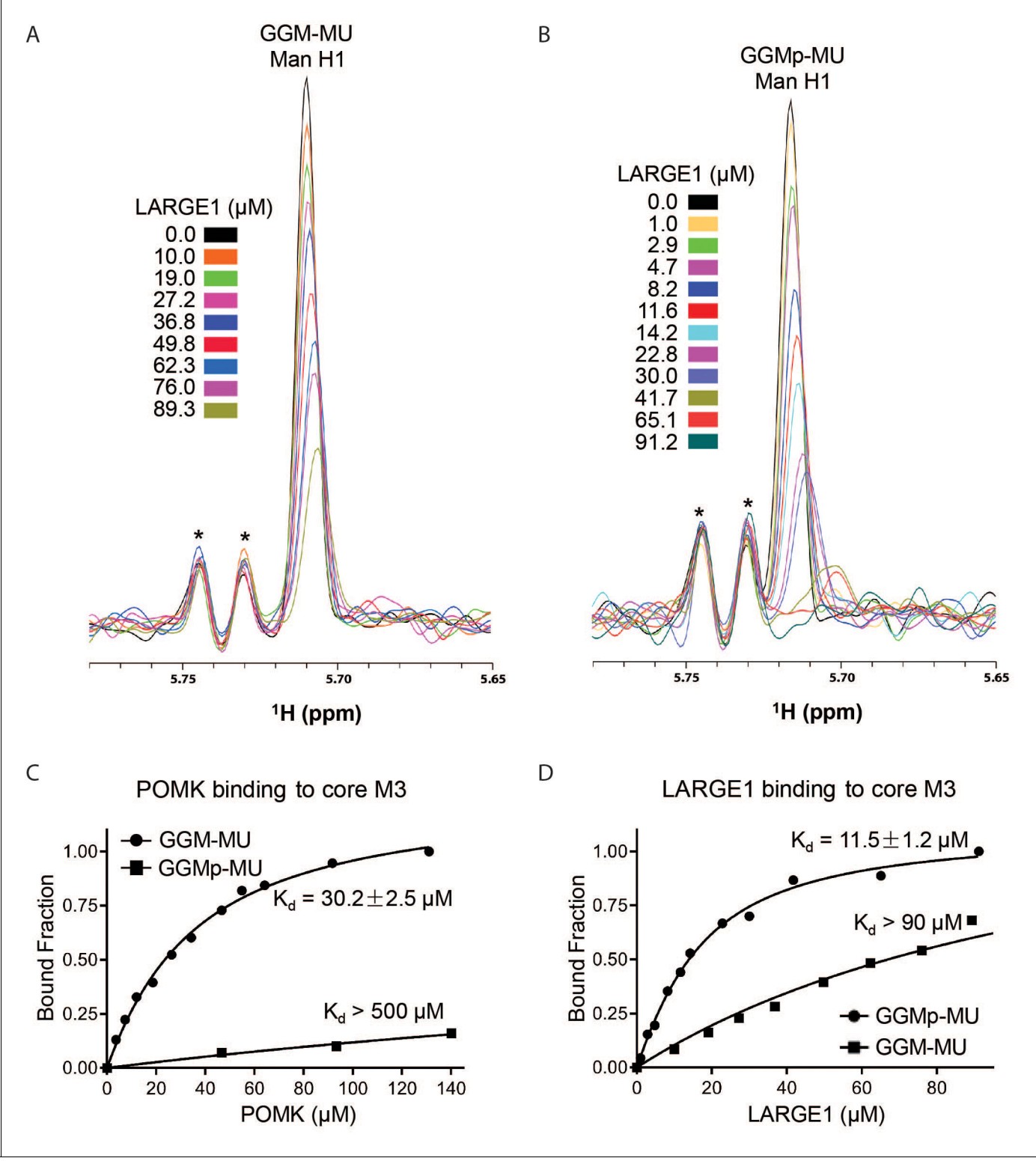

**Figure 8.** NMR Analyses of POMK and LARGE1 Binding to GGM-MU and GGMp-MU. (**A, B**) 1D $^1$H NMR spectra of the anomeric region of GGM-MU (**A**) and GGMp-MU (**B**) were acquired for the glycan concentration of 10.0 µM in the presence of various concentrations of LARGE1 as indicated. The peak Man H1 is derived from the mannose anomeric H1 proton. Stars indicate impurity peaks derived from buffer. (**C, D**) Fitting of the NMR binding data of POMK (**C**) and LARGE1 (**D**) to core M3 glycans of GGM-MU and GGMp-MU, respectively. The bound fraction was obtained from the NMR titration data by measuring the difference in the peak intensity of the anomeric proton Man H1 in the absence (free form) and presence (bound form) of POMK or LARGE1, then divided by the peak intensity of the free form.

*Figure 8 continued on next page*

*Figure 8 continued*

The online version of this article includes the following figure supplement(s) for figure 8:

**Figure supplement 1.** NMR Spectra of POMK Binding to GGMp-MU and Structure of GGMp-MU.

**Figure supplement 2.** Model of Full-Length and Non-extended Matriglycan Synthesis.

**Figure supplement 3.** Biochemical and Histologic Analysis of $Mck^{Cre}$; $Pomk^{LoxP/LoxP}$ Quadriceps Muscle.

prevent eccentric contraction-induced force loss or skeletal muscle pathology. Interestingly, missense mutations in FKRP that cause LGMD2I also show reduced expression of matriglycan (*Yoshida-Moriguchi and Campbell, 2015*) and exhibit a milder muscular dystrophy. Thus, M-POMK KO mice are an excellent model of milder forms of dystroglycanopathy in which short matriglycan is expressed and will be useful for future studies of these forms of dystroglycanopathy.

α-DG is composed of three domains: the DGN, which undergoes cleavage at arginine-312 by a furin-like convertase during α-DG post-translational processing, a central mucin-like domain, and a C-terminus (*Kanagawa et al., 2004*; *Singh et al., 2004*). The natural C-terminal domain boundary of DGN, arginine-312 in humans, is proximal to three sites of matriglycan synthesis (threonines-317, 319, 379) within the mucin-like domain of α-DG. Biochemical studies using various *POMK* KO HAP1 cell lines demonstrated that the synthesis of the short, non-extended form of matriglycan occurs on threonine-317 of the mucin-like domain and, like full-length matriglycan, requires LARGE1, DGN, and CDP-ribitol. Cell biological experiments demonstrated that the DGN is necessary for synthesis of the short form of matriglycan. As the binding of LARGE1 to the DGN is essential for the synthesis of full-length matriglycan on α-DG (*Kanagawa et al., 2004*; *Hara et al., 2011a*), it is required for synthesis of the short form of matriglycan as well. Solution NMR studies revealed that LARGE1 binds to core M3, and the binding affinity increases in the presence of the mannose phosphate. The phosphorylated core M3, could, therefore serve to recruit DGN-bound LARGE1 to the proper residue during the initiation of full-length matriglycan synthesis. In the absence of the mannose phosphate, the DGN-bound LARGE1 may instead act only upon the matriglycan acceptor added to threonine-317, the threonine nearest to the DGN. Synthesis of full-length matriglycan may, therefore, proceed through a complex of DGN, LARGE1, and phosphorylated core M3. The phosphorylated core M3 may also serve to anchor LARGE1 to α-DG during matriglycan elongation. In the absence of POMK, the binding of LARGE1 to the DGN and the unphosphorylated core M3 may only be sufficient for synthesis of a short form of matriglycan. Further structural and biochemical studies will be required to understand the precise interactions between DGN, LARGE1, and the phosphorylated core M3. Taken together, our results indicate that LARGE1 requires DGN to synthesize the short, non-extended form of matriglycan but needs both the DGN and the phosphorylated core M3 to generate full-length matriglycan on α-DG.

Our study demonstrates that POMK is required for the synthesis of full-length and high-molecular weight forms of matriglycan (*Figure 8—figure supplement 2A*). In the absence of POMK, LARGE1 generates a short, non-extended form of matriglycan (*Figure 8—figure supplement 2B*). Collectively, our work provides the first insights into the pathogenic mechanism behind POMK-deficient muscular dystrophy and better elucidates how full-length matriglycan is synthesized so it can act as a scaffold for ECM proteins, thereby enabling proper skeletal muscle function and preventing muscular dystrophy.

## Materials and methods

### Patient information

Patient NH13-284 received a diagnosis of congenital muscular dystrophy (CMD) with brain malformations.

### Generation of $Pomk^{LoxP/LoxP}$ mice

The *Pomk* gene consists of five exons, exons 1, 2, and 3, which are non-coding and exons 4 and 5, which are coding (*Zhu et al., 2016*; *Di Costanzo et al., 2014*). We used Clustered Regularly Interspersed Short Palindromic Repeats (CRISPR)-Cas9 to insert LoxP sites around exon 5.

*Pomk*_5P1 TTCTTTCTGTGATGTGTGCTTATTC
*Pomk*_5P2 CAGACACTCACCCTTTACCTTAG
Wild-type: 197 bp
Targeted: 235 bp
*Pomk*_3P1 AGCCACACCTTCCTACAGTC
*Pomk*_3P2 AAGCTCTGCCCAGAGAGAAG
Wild-type: 123 bp
Targeted: 162 bp
*Pomk*_5'_guide(601) CGTGTCCCGCCAGGAATGAA
*Pomk*_3'_guide(3P1) TCAGGAGGCGGCTCCCAGTG
*Pomk*_5'_donor(601; PAGE purified)
TCCTCATCTTCTCCCTGTGCAGTCAATCTGCACAGCTCCCTGCACACATGGCTTATAGAG
TGGTTCTCACCCCGCCCTTCATAACTTCGTATAGCATACATTATACGAAGTTATGGTACCTCC
TGGCGGGACACGAATAAGCACACATCACAGAAAGAAGTCTGTTGTCTTGAC
TGCCCAGCCCTCCGCAGCTGCCACCC
*Pomk*_3'_donor(3P1; PAGE purified)
AGTGTGAGATTCAAGTGTGGATATGCAGTGATCCTCTGGCCACACTTGTGAGCAGCCA-
CACCTTCCTACAGTCCCTCACTATAACTTCGTATAGCATACATTATACGAAGTTATGGA
TCCGGGAGCCGCCTCCTGAGCCCTGCTGTGTAACCCACCTACCTTCCCTCCTTTCACAC
TAGAAGCTGAGAGCTCTTCTCTTC

## Animals

B6SJLF1/J mice were purchased from Jackson Labs (100012; Bar Harbor, ME). Male mice older than 8 weeks were used to breed with 3–5 week-old super-ovulated females to produce zygotes for electroporation. Female ICR (Envigo, Indianapolis, IN; Hsc:ICR(CD-1)) mice were used as recipients for embryo transfer.

Mice expressing *Cre* under the *mouse creatine kinase* (*Mck*) promoter, B6.FVB(129S4)-Tg(Ckmm-cre)5Khn/J (stock no. 006475) (*Brüning et al., 1998*) and the *Pax7* promoter, *Pax7*^tm1(cre)Mrc^/J, (stock no. 010530) (*Keller et al., 2004*) were purchased from the Jackson Laboratory. Male mice expressing the *Mck-Cre* transgene were bred to female mice homozygous for the floxed *Pomk* allele (*Pomk*^LoxP/LoxP^). Male F1 progeny with the genotype *Mck*^Cre^; *Pomk*^LoxP/+^ were bred to female *Pomk*^LoxP/LoxP^ mice. A *Cre* PCR genotyping protocol was used to genotype the *Cre* allele using standard *Cre* primers. The primers used were Sense: TGATGAGGTTCGCAAGAACC and Antisense: CCATGAG TGAACGAACCTGG.

Sanger sequencing of tail DNA was performed by the University of Iowa Genome Editing Core Facility to confirm incorporation of 5' and 3' LoxP sites. PCR probes were developed at Transnetyx to genotype mice expressing both *Pax7-Cre* and *Mck-Cre*. Genotyping of *Mck*^Cre^; *Pax7*^Cre^; *Pomk*^LoxP/LoxP^ mice was performed by Transnetyx using real-time PCR.

All mice were socially housed in a barrier-free, specific pathogen-free conditions as approved by the University of Iowa Animal Care and Use Committee (IACUC). All animals were maintained in a climate-controlled environment at 25°C and a 12/12 light/dark cycle. Animal care, ethical usage, and procedures were approved and performed in accordance with the standards set forth by the National Institutes of Health and IACUC. For studies with *Mck*^Cre^; *Pomk*^LoxP/LoxP^ mice, N = 3 mice of each genotype (*Pomk*^LoxP/LoxP^ and *Mck*^Cre^; *Pomk*^LoxP/LoxP^) were used. For studies with *Mck*^Cre^; *Pax7*^Cre^; *Pomk*^LoxP/LoxP^ mice, animals of varying ages were used as indicated, and N = 3 each of *Pomk*^LoxP/LoxP^ and *Mck*^Cre^; *Pax7*^Cre^; *Pomk*^LoxP/LoxP^ were used. Littermate controls were employed whenever possible. The number of animals required was based on previous studies (*de Greef et al., 2016*; *Goddeeris et al., 2013*) and experience with standard deviations of the given techniques.

## Preparation of Cas9 RNPs and the microinjection mix

Chemically modified CRISPR-Cas9 crRNAs and CRISPR-Cas9 tracrRNAs were purchased from Integrated DNA Technologies (IDT) (Alt-R CRISPR-Cas9 crRNA; Alt-R CRISPR-Cas9 tracrRNA (Cat# 1072532)). The crRNAs and tracrRNA were suspended in T10E0.1 and combined to 1 µg/µL (~29.5 µM) final concentration in a 1:2 (µg: µg) ratio. The RNAs were heated at 98°C for 2 min and allowed to cool slowly to 20°C in a thermal cycler. The annealed cr:tracrRNAs were aliquoted to single-use tubes and stored at −80°C.

Cas9 nuclease was also purchased from IDT (Alt-R S.p. HiFi Cas9 Nuclease). Cr:tracr:Cas9 ribonucleoprotein complexes were made by combining Cas9 protein and each cr:tracrRNA; final concentrations: 60 ng/µL (~0.4 µM) Cas9 protein and 60 ng/µL (~1.7 µM) cr:tracrRNA). The Cas9 protein and annealed RNAs were incubated at 37°C for 10 min. The two RNP mixes were combined and incubated at 37°C for an additional 5 min. The single stranded oligonucleotide donors (ssODN) were purchased from IDT as Ultramers. The ssODNs were added to the RNPs and the volume adjusted to the final concentrations in the injection mix were 10 ng/µL each ssODN; 20 ng/µL each guide RNA and 40 ng/µL Cas9 protein.

## Collection of embryos and microinjection

Pronuclear-stage embryos were collected using previously described methods (*Pinkert, 2002*). Embryos were collected in KSOM media (Millipore, Burlington, MA; MR101D) and washed three times to remove cumulous cells. Cas9 RNPs and ssODNs were injected into the pronuclei of the collected zygotes and incubated in KSOM with amino acids at 37°C under 5% $CO_2$ until all zygotes were injected. Fifteen to 25 embryos were immediately implanted into the oviducts of pseudo-pregnant ICR females.

Insertion of loxP1 (5') and loxP2 (3') sites was confirmed by cloning and sequencing of genomic PCR products (*Figure 3—figure supplement 2*) from tail DNA of filial 0 (F0) *Pomk*[LoxP/+] mice using primers flanking the 5' loxP site, ACTCCAGTTGGTTTCAGGAAG and GAGGGAAGAGAAGTCAGGAAAG. For the 3' loxP site, primers of sequence ACCGAGTGTGAGATTCAAGTG and GGTTGCTGGTAGGGTTAAGAG were used. The 5' loxP site contains a *Kpn*I cleavage site, and the 3' loxP site contains a *Bam*HI site. The screen of the 5' loxP site gives a product of 803 base pairs for the *LoxP* allele when uncut. *Kpn*I digestion of the 5' loxP site gives three products of 381, 355, and 67 base pairs. A screen of the 3' loxP site gives a product of 396 base pairs for the uncut allele with loxP site, while *Bam*HI digestion of the 3' loxP site gives products of 273 and 123 base pairs.

Genotyping was carried out using primers flanking the exon five loxP1 site or the (TTCTTTCTGTGATGTGTGCTTATTC) or loxP2 (CAGACACTCACCCTTTACCTTAG) site. The wild-type allele is 197 bp while the floxed allele is 235 bp. *Pomk*[LoxP/+] mice were backcrossed five generations onto a C57BL/6J background and backcrossed mice used whenever possible.

## Forelimb grip strength test

Forelimb grip strength was measured at 1 month and 4 months of age using previously published methods (*de Greef et al., 2016*). A mouse grip strength meter (Columbus Instruments, Columbus, OH) was mounted horizontally, with a nonflexible grid connected to the force transducer. The mouse was allowed to grasp the grid with its two front paws and then pulled away from the grid by its tail until the grip was broken. This was done three times over five trials, with a one-minute break between each trial. The gram force was recorded per pull, and any pull where only one front limb or any hind limbs were used were discarded. If the mouse turned, the pull was also discarded. After 15 pulls (5 sets of 3 pulls), the mean of the three highest pulls of the 15 was calculated and reported. Statistics were calculated using GraphPad Prism eight software. Student's t-test was used (two-sided). Differences were considered significant at a p-value less than 0.05. Graph images were also created using GraphPad Prism and the data in the present study are shown as the means + / - SD unless otherwise indicated.

## Creatine kinase assay

Creatine kinase levels were measured in 8-week-old mice 2 hr after mild downhill run (three meters per minute for 5 min followed by 15 m per minute for 10 min) at a 15- degree downhill incline as previously described (*de Greef et al., 2016*; *Goddeeris et al., 2013*). Blood was collected by tail vein bleeds from non-anesthetized, restrained mice using a Microvette CB300 (Sarstedt AG and Co, Newton, NC). Samples were centrifuged at 12,000 rpm for 10 min and prepared using an enzyme-coupled CK kit (Stanbio Laboratory, Boerne, TX) using the manufacturer's instructions. Absorbance was measured using a plate reader at 340 nm every 30 s for 2 min at 37°C. Statistics were calculated using GraphPad Prism software and Student's t-test was used (two-sided). Differences were considered significant at a p-value less than 0.05. Graph images were also created using GraphPad Prism eight and the data in the present study are shown as the means + / - SD unless otherwise indicated.

## Body weight measurements

Mice were weighed as previously described (*de Greef et al., 2016*). Weights were measured after testing grip strength using a Scout SPX222 scale (OHAUS Corporation, Parsippany, NJ), and the tester was blinded to genotype. Statistics were calculated using GraphPad Prism eight software and Student's t-test was used (two-sided). Differences were considered significant at a p-value less than 0.05. Graph images were also created using GraphPad Prism and the data in the present study are shown as the means + / - SD unless otherwise indicated.

## Measurement of *in vitro* muscle function

To compare the contractile properties of muscles, extensor digitorum longus (EDL) muscles were surgically removed as described previously (*Rader et al., 2016*; *de Greef et al., 2016*). The muscle was immediately placed in a bath containing a buffered physiological salt solution (composition in mM: NaCl, 137; KCl, 5; $CaCl_2$, 2; $MgSO_4$, 1; $NaH_2PO_4$, 1; $NaHCO_3$, 24; glucose, 11). The bath was maintained at 25°C, and the solution was bubbled with 95% $O_2$ and 5% $CO_2$ to stabilize pH at 7.4. The proximal tendon was clamped to a post and the distal tendon tied to a dual mode servomotor (Model 305C; Aurora Scientific, Aurora, ON, Canada). Optimal current and whole muscle length ($L_0$) were determined by monitoring isometric twitch force. Optimal frequency and maximal isometric tetanic force ($F_0$) were also determined. The muscle was then subjected to an eccentric contraction (ECC) protocol consisting of eight eccentric contractions (ECCs) at 3 min intervals. A fiber length $L_f$-to-$L_0$ ratio of 0.45 was used to calculate $L_f$. Each ECC consisted of an initial 100 millisecond isometric contraction at optimal frequency immediately followed by a stretch of $L_o$ to 30% of $L_f$ beyond $L_o$ at a velocity of 1 $L_f$/s at optimal frequency. The muscle was then passively returned to $L_o$ at the same velocity. At 3, 15, 30, 45, and 60 min after the ECC protocol, isometric tetanic force was measured. After the analysis of the contractile properties, the muscle was weighed. The CSA of muscle was determined by dividing the muscle mass by the product of $L_f$ and the density of mammalian skeletal muscle (1.06 g/cm$^3$). The specific force was determined by dividing $F_o$ by the CSA (kN/mm$^2$). 18–20-week-old male mice were used, and right and left EDL muscles from each mouse were employed whenever possible, with n = 5 to 8 muscles used for each analysis. Each data point represents an individual EDL. Statistics were calculated using GraphPad Prism eight software and Student's unpaired t-test was used (two-sided). Differences were considered significant at a p-value less than 0.05.

## H&E and immunofluorescence analysis of skeletal muscle

Histology and immunofluorescence of mouse skeletal muscle were performed as described previously (*Goddeeris et al., 2013*). Mice were euthanized by cervical dislocation and directly after sacrifice, quadriceps muscles were isolated, embedded in OCT compound and then snap frozen in liquid nitrogen-cooled 2-methylbutane. 10 µM sections were cut with a cryostat (Leica CM3050S Research Cryostat; Amsterdam, the Netherlands) and H&E stained using conventional methods. Whole digital images of H&E-stained sections were taken by a VS120-S5-FL Olympus slide scanner microscope (Olympus Corporation, Tokyo, Japan). For immunofluorescence analyses, a mouse monoclonal antibody to glycoepitopes on the sugar chain of α-DG (IIH6, 1:100 dilution, Developmental Studies Hybridoma Bank, University of Iowa; RRID:AB_2617216) was added to sections overnight at 4°C followed by Alexa Fluor-conjugated goat IgG against mouse IgM (Invitrogen, Carlsbad, CA, 1:500 dilution), for 40 min. The sections were also stained with rabbit polyclonal antibody to β-DG (AP83; 1:50 dilution) followed by Alexa Fluor-conjugated 488 Goat anti-rabbit IgG (1:500). Whole sections were imaged with a VS120-S5-FL Olympus slide scanner microscope. Antibody IIH6 is a monoclonal to the glycoepitope of α-DG (*Ervasti and Campbell, 1991*), and AP83 is a polyclonal antibody to the C-terminus of β-DG (*Ervasti and Campbell, 1991*), both of which have been described previously.

For histologic analysis of human skeletal muscle, H&E staining on 10 µm frozen section was performed using the Leica ST5020 Multistainer workstation (Leica Biosystems, Buffalo Grove, IL) according manufacturer's instructions. For immunofluorescence analysis, unfixed frozen serial sections (7 µm) were incubated with primary antibodies for 1 hr, and then with the appropriate biotinylated secondary antibodies for 30 min followed by streptavidin conjugated to Alexa Fluor 594 (ThermoFisher Scientific, UK) for 15 min. Primary antibodies used were mouse monoclonal: α-DG IIH6 (clone IIH6C4) (*Ervasti and Campbell, 1991*), β-DG (Leica, Milton Keynes, UK; clone 43DAG1/8D5). All

washes were made in PBS and incubations were performed at room temperature. Sections were evaluated with a Leica DMR microscope interfaced to MetaMorph (Molecular Devices, Sunnyvale, CA).

## Tissue biochemical analysis

30 slices of 30 µM thickness were taken with a cryostat (Leica CM3050S Research Cryostat) from skeletal muscle or heart that had been frozen in liquid nitrogen-cooled 2-methylbutane. For biochemical analysis of murine skeletal muscle, quadriceps muscle were used.

Samples were solubilized in 500 µL of 1% Triton X-100 in 50 mM Tris pH 7.6 and 150 mM NaCl with protease inhibitors (per 10 mL buffer: 67 µL each of 0.2 M phenylmethylsulfonylfluoride (PMSF), 0.1 M benzamidine and 5 µL of each of leupeptin (Sigma/Millipore) 5 mg/mL, pepstatin A (Millipore) 1 mg/mL in methanol, aprotinin (Sigma-Aldrich) 5 mg/mL, calpeptin (Fisher/EMD Millipore) 1.92 mg/mL in Dimethyl Sulfoxide (DMSO), Calpain Inhibitor 1 (Sigma-Aldrich) 1.92 mg/mL in DMSO). Samples were vortexed for 4 min and solubilized for 2.5 hr at 4°C with rotation. Samples were then spun down at 12,000 rpm for 30 min at 4°C on a Beckman Tabletop Centrifuge. The supernatant was incubated with 100 µL WGA-Agarose slurry (Vector Biolabs, Malvern, PA, AL-1023) overnight at 4°C with rotation. The next day samples were washed three times in 50 mM Tris pH 7.6 and 150 mM NaCl with 0.1% TX-100 and protease inhibitors. 100 µL of 5X Laemmli Sample Buffer (LSB) was added, samples boiled for 10 min, and 125 µL of this was loaded in each lane of gels for western blotting.

## Fibroblast growth and flow cytometry

Fibroblasts used for biochemical analyses were grown in 20% Fetal Bovine Serum (FBS, Life Technologies, Carlsbad, CA) and 1% penicillin/streptomycin (Invitrogen). Cells were split at 1:2 every 2 days using Trypsin-EDTA (ThermoFisher Scientific, Waltham, MA).

For flow cytometry analyses, fibroblasts cultured from skin biopsies were grown in Dulbecco's modified Eagles medium (Invitrogen) with 20% fetal bovine serum (FBS, Life Technologies), 1% glutamax (Thermo Fisher Scientific) and 1% penicillin/streptomycin (Sigma-Aldrich). Upon approximately 90% confluence, cells were washed with PBS without Ca and Mg, detached with non-enzymatic dissociation solution (Sigma-Aldrich cat. C5914) and fixed in 2% paraformaldehyde for 10 min. Cells were subsequently incubated on ice with the following antibodies diluted in PBS/0.1% FBS: anti-α-DG IIH6 (Millipore) for 30 min, anti-mouse biotinylated IgM (Vector Labs, Burlingame, CA) for 20 min, Streptavidin-Phycoerythrin (BD Pharmingen) for 15 min. Negative controls for each fibroblast population were incubated with 0.1% FBS/PBS without the primary antibodies. Cells were washed twice and centrifuged at 1850 g for 4 min, after each incubation step. After the last wash, cell pellets were resuspended in 500 µL of PBS. A total of 10,000 event were acquired using the Cyan ADP analyser (Beckman Coulter, Brea, CA) and analysed using FlowJo software version 7.6.5 (Tree Star, USA).

## Generation and characterization of HAP1 mutant cell lines

HAP1 cells (RRID:CVCL_Y019) are a haploid human cell line with an adherent, fibroblast-like morphology, originally derived from parent cell line KBM-7 (RRID:CVCL_A426). WT C631 cells (a diploid cell line containing duplicated chromosomes of HAP1) were purchased from Horizon Discovery, and gene-specific knockout (KO) HAP1 cells were generated by Horizon Discovery. Absence of the gene was confirmed via PCR amplification and Sanger sequencing. The identity of the cells has been authenticated by the company using the STR profiling method. Mycoplasma testing of the cells was performed on a routine basis to ensure the cells are not contaminated. HAP1 KO cell lines have complete loss of gene function and are validated in the lab by performing western blot analysis before and after gene transfer with the appropriate gene. For each HAP1 KO cell line, a matched WT control parental cell line (WT C631) was provided, ensuring that phenotypes can be attributed directly to the genetic modification. These cells are cultured in Iscove's Modified Dulbecco's medium (IMDM) supplemented with 10% fetal calf serum and 1% Pen-Strep antibiotics.

### *POMK* knockout (KO) HAP1

HAP1 cells bearing a 10 bp deletion of exon 4 of *POMK*, generated using the CRISPR/Cas9 system, were purchased from Horizon Discovery (HZGHC001338c004, clone 1338–4) and were previously

described (*Zhu et al., 2016*). *POMK* knockout (KO) HAP1 cells lack the single copy of the wild-type *POMK* allele and are therefore null at the *POMK* locus. The sequence of the guide RNA used is TGA-GACAGCTGAAGCGTGTT. Absence of the wild-type *POMK* allele was confirmed by Horizon Discovery via PCR amplification and Sanger sequencing. PCR primers used for DNA sequencing are *POMK* Forward 5'-ACTTCTTCATCGCTCCTCGACAA-3', and *POMK* Backward 5'- GGATGCCACACTGC TTCCCTAA-3'. The identity of the cells has been authenticated by the company using the STR profiling method. Mycoplasma testing of the cells was performed on a routine basis to ensure the cells are not contaminated.

### *POMK/DAG1* KO HAP1

HAP1 cells lacking both *POMK* and *DAG1* expression (*POMK/DAG1* KO HAP1 cells) were generated using CRISPR/Cas9 by Horizon Discovery. A 16 bp deletion in the DAG1 gene (exon 2) was introduced into the *POMK* KO HAP1 line (HZGHC001338c004). The sequence of the guide RNA is CCGACGACAGCCGTGCCATC; NM_004393. PCR primers for DNA sequencing were forward TAG-CAAGACTATCGACTTGAGCAAA and reverse GCAATCAAATCTGTTGGAATGGTCA.

### *POMK/LARGE1* KO HAP1

HAP1 cells lacking both *POMK* and *LARGE1* expression (*POMK/LARGE1* KO HAP1 cells) (HZGHC007364c011) were generated using CRISPR/Cas9 by Horizon Discovery. A 43 bp deletion of exon 3 of *LARGE1* was introduced into the *POMK* KO HAP1 line (HZGHC001338c004). The guide RNA sequence was CTCGGCGATGGGATGGGGCT and the primer sequence was PCR forward GAGGCATGGTTCATCCAGATTAAAG and PCR reverse CTTTACCTCGCATTTCTCCACGA.

### *POMK/ISPD* KO HAP1

HAP1 cells containing a 1 bp insertion of exon 4 of the *POMK* gene, generated using the CRISPR/Cas9 system, were purchased from Horizon Discovery (HZGHC001338c001, clone 1338–1). The mutation in *POMK* is predicted to lead to a frameshift. These cells also lacked expression of *ISPD*. The guide RNA sequence was TGAGACAGCTGAAGCGTGTT. The sequences of PCR primers were PCR forward ACTTCTTCATCGCTCCTCGACAA and PCR reverse GGATGCCACACTGCTTCCCTAA.

### *LARGE1* KO HAP1

HAP1 cells (clone 122–6, HZGHC000122c006) were purchased from Horizon Discovery. Cells were generated using a CRISPR/Cas9-mediated 1 bp deletion of exon 3 of *LARGE*. The guide RNA sequence was GCTCTCGCGCTCCCGCTGGC and the primer sequence for 122–7 was PCR forward ATGGAGTAGGTCTTGGAGTGGTT and PCR reverse GAGGCATGGTTCATCCAGAGTTAAAG.

### *FKTN* KO HAP1

HAP1 cells (clone 721–10, catalog number 32597–10) were purchased from Horizon Discovery. CRISPR/Cas9 was used to introduce a 16 bp deletion of exon 3 of *FKTN*. The sequence of the guide RNA was CAGAACTTGTCAGCGTTAAA and the sequences of PCR forward CAGATCAAAGAA TGCCTGTGGAAAT and PCR reverse TGCAAAGAGAAGTGTGATCAGAAAA.

## Adenovirus production

DGE (Delta H30- A316) was generated and described previously (*Hara et al., 2011a*; *Kanagawa et al., 2004*; *Kunz et al., 2001*). DG T317A, DG T319A, and DG T317A/T319A were first subcloned into an Fc-tagged DG construct (DGFc3) (*Hara et al., 2011b*; *Kanagawa et al., 2004*; *Kunz et al., 2001*). The *Kpn*I-*Xho*I fragments from the DGFc3 mutants corresponding to the mutant constructs (DG T317A, DG T319A, or DG T317A/T319A) were then subcloned into pAd5RSVK-NpA (obtained from the University of Iowa Viral Vector Core) as was the *Xho*I-*Xba*I fragment from an adenovirus encoding dystroglycan WT. *E1*-deficient recombinant adenoviruses (Ad5 RSV DG WT, DG T317/T319, DG T317A, DG T319A, DGE, Ad-POMK WT) were generated by the University of Iowa Viral Vector Core (VVC) using the RAPAd system (*Anderson et al., 2000*). Assays for replication competence of adenoviruses were performed to check for contamination. Ad-POMK WT and Ad-POMK D204A were generated by ViraQuest Inc (North Liberty, IA) using the RAPAd system and was described previously (*Zhu et al., 2016*). Ad-POMK D204N was also generated by ViraQuest Inc

Absence of the viral *E1* DNA sequence was confirmed by ViraQuest Inc after PCR amplification of the viral DNA and staining on DNA agarose gel electrophoresis. Replication competence of adenoviruses was negative as assessed by plaque forming assays in cells performed from $10^9$ viral particles up to 14 days. Adenoviral Fukutin (FKTN) and ISPD have been described previously (*Willer et al., 2012*). Adenoviral LARGE1 has been described previously (*Barresi et al., 2004*). DGFc340TEV was cloned into the pUC57-mini vector by GenScript (*Hara et al., 2011a*; *Kanagawa et al., 2004*; *Kunz et al., 2001*). The insert includes TEV protein cleavage site between amino acids (AAs) 1–340 of rabbit DG and human IgG1 Fc. The insert was subcloned in pcDNA3 expression vector with *Eco*RI. Subsequently, *Fse*I-x-340 AAs DG-TEV-6xHis-*Not*I fragment was obtained using pcDNA3DGFc340-TEV as a PCR template. *Fse*I-x-340 AAs DG-TEV-6xHis-*Not*I was ligated into pcDNA3DGFc340TEV digested with *Fse*I and *Not*I to construct DG340TEVHis, which includes 1–340 AAs of rabbit DG, TEV site, and 6x Histidine. The construct was also inserted in pacAd shuttle plasmid from the VVC to generate the adenoviral vector. Next, *Fse*I-x-390 AAs-TEV-6xHis-*Not*I was obtained using pcDNA3rbtDG as a PCR template and ligated into the pcDNA3DG340TEVHis digested with *Fse*I and *Not*I to construct DG390TEVHis, which includes 1–390 AAs of rabbit DG, TEV site, and 6x Histidine. The construct was also inserted in pacAd shuttle plasmid from the VVC to generate the Ad virus vector. *E1*-deficient recombinant adenoviruses were generated by the University of Iowa Viral Vector Core using the RAPAd system (*Kunz et al., 2001*).

## HAP1 cell culture and adenovirus infection

HAP1 cells were maintained at 37°C and 5% $CO_2$ in Iscove's Modified Dulbecco's Medium (IMDM, Gibco) supplemented with 10% Fetal Bovine Serum (FBS) and 1% penicillin/streptomycin (Invitrogen). Cells were split every 3 days at 1:10 using Trypsin-EDTA (ThermoFisher Scientific). On day one for adenovirus transfection experiments, media was changed to 2% IMDM, and an average of $5.9 \times 10^6$ *POMK* KO HAP1 cells were infected at the indicated multiplicity of infection (MOI) with the indicated adenovirus. On day 2, infection medium was replaced with 10% IMDM, and on day three the cells were processed for biochemical analyses.

## Glycoprotein isolation and biochemical analyses from cultured cells

For western blots and laminin overlay, HAP1 cells and fibroblasts were washed twice in ice-cold Dulbecco's phosphate-buffered saline (DPBS, Gibco). The second PBS wash contained the protease inhibitors (0.23 mM PMSF and 0.64 mM benzamidine). Plates were scraped, spun down for 5 min at 14, 000 rpm at 4°C, and pellets were solubilized in 1% Triton X-100 in Tris-buffered saline (TBS, 50 mM Tris-HCl pH 7.6, 150 mM NaCl) with protease inhibitors (0.23 mM PMSF and 0.64 mM benzamidine) for 1 hr at 4°C. Samples were then spun down at 14,000 rpm for 5 min, and supernatants incubated in 200 µL wheat-germ agglutinin (WGA)-agarose (Vector Laboratories, AL-1023) as previously described (*Michele et al., 2002*; *Goddeeris et al., 2013*). The following day, WGA beads were washed three times with 0.1% Triton X-100-TBS plus protease inhibitors and heated to 99°C for 10 min with 250 µL of 5X Laemmli sample buffer. Samples were run on SDS-PAGE and transferred to PVDF-FL membranes (Millipore) as previously published (*Michele et al., 2002*; *Goddeeris et al., 2013*).

## Immunoblotting and ligand overlay

The mouse monoclonal antibody against α-DG (IIH6, Developmental Studies Hybridoma Bank, University of Iowa; RRID:AB_2617216) was characterized previously and used at 1:100 (*Ervasti and Campbell, 1991*). The polyclonal antibody, AF6868 (R and D Systems, Minneapolis, MN; RRID:AB_10891298), was used at a concentration of 1:200 for immunoblotting the core α-DG and β-DG proteins, and the secondary was a donkey anti-sheep (LI-COR Bioscience, Lincoln, NE) used at 1:2000 concentration. Anti-POMK (Novus Biologicals, Littleton, CO, 6f10) was used at 1:500, and the secondary was 1:2000 Goat anti-Mouse IgG1 (LI-COR Bioscience). The antibody against the Na/K ATPase (BD Biosciences, San Jose, CA, 610993) was used at 1:1000 in 5%-milk Blotto, and the secondary was 1:10,000 Goat anti-Mouse IgG1 (LI-COR Bioscience). Anti-myc (Millipore Sigma, Clone 4A6) was used at 1:2000 in 2% milk and the secondary was 1:2000 Goat anti-Mouse IgG1 (LI-COR Bioscience). Blots were developed with infrared (IR) dye-conjugated secondary antibodies (LI-COR

Bioscience) and scanned using the Odyssey infrared imaging system (LI-COR Bioscience). Blot images were captured using the included Odyssey image-analysis software.

Laminin overlay assays were performed as previously described (*Michele et al., 2002*; *Goddeeris et al., 2013*). PVDF-FL membranes were blocked in laminin-binding buffer (LBB: 10 mM triethanolamine, 140 mM NaCl, 1 mM MgCl$_2$, 1 mM CaCl$_2$, pH 7.6) containing 5% milk followed by incubation with mouse Engelbreth-Holm-Swarm (EHS) laminin (ThermoFisher, 23017015) overnight at a concentration of 7.5 nM at 4°C in LBB containing 3% bovine serum albumin (BSA) and 2 mM CaCl$_2$. Membranes were washed and incubated with anti-laminin antibody (L9393; Sigma-Aldrich 1:1000 dilution) followed by IRDye 800 CW dye-conjugated donkey anti-rabbit IgG (LI-COR, 926–32213) at 1:2500 dilution.

## EDTA treatment of ligand overlays

EDTA treatment of laminin overlay assays was performed as described above for laminin overlays; however, calcium was excluded from all buffers made with LBB (i.e. 5% milk-LBB, 3% BSA-LBB) and 10 mM EDTA was added to all LBB-based buffers, including LBB wash buffer, 5% milk-LBB, and 3% BSA-LBB buffers.

## POMK assay

HAP1 cells were washed twice in ice-cold PBS, scraped, and spun down at 14,000 rpm for 5 min at 4°C. After removing supernatant, the cell pellet was resuspended in 0.1 M MES buffer pH 6.5 with 1% Triton X-100 with Protease Inhibitors (0.23 mM PMSF and 0.64 mM Benzamidine) for 1 hr at 4°C rotating. Samples were spun down again, and the supernatant was incubated with 200 μL of WGA-agarose beads (Vector Biolabs, AL-1023) overnight at 4°C with rotation. Samples were washed the next day three times in 0.1 M MES pH 6.5 with 0.1% Triton X-100 and protease inhibitors, and 100 μL of the beads were resuspended in 100 μL of the wash buffer.

For fibroblast POMK activity measurements, cells were processed as above and solubilized in 1% TX-100 in 50 mM Tris and 150 mM NaCl pH 7.6 with protease inhibitors as described above and incubated with WGA-agarose beads. The next day, WGA beads were washed three times and resuspended in 0.1% TX-100 in 0.1 M MES pH 6.5 buffer with protease inhibitors.

For measurement of mouse and human skeletal muscle POMK activity, 30 slices of 30 μM thickness were taken using a Leica 3050 s cryostat from quadriceps muscle frozen in liquid nitrogen-cooled 2-methylbutane. Samples were solubilized in 250 μL of 1% Triton X-100 in 0.1 M MES pH 6.5 with protease inhibitors (per 10 mL buffer: 67 μL each of 0.2 M PMSF, 0.1 M benzamidine and 5 μL/10 mL of buffer of leupeptin (Sigma/Millipore) 5 mg/mL, pepstatin A (Millipore) 1 mg/mL in methanol, aprotinin (Sigma-Aldrich) 5 mg/mL, calpeptin (Fisher/EMD Millipore) 1.92 mg/mL in dimethyl sulfoxide (DMSO), calpain inhibitor 1 (Sigma-Aldrich) 1.92 mg/mL in DMSO). Samples were solubilized for 2.5 hr at 4°C on a rotator. Samples were then spun down at 14,000 rpm for 30 min at 4°C on a Beckman Tabletop Centrifuge. The supernatant (total lysate) was separated from the pellet, and 10 μL of this was used for POMK assays.

For POMK reaction in HAP1 cells and fibroblasts, 20 μL slurry (consisting of 10 μL beads and 10 μL MES buffer) was incubated with reaction buffer for a final reaction volume of 40 μL. For POMK assay from skeletal muscle, 10 μL of total lysate was incubated with 20 μL of reaction buffer for a reaction volume of 30 μL. The final reaction concentration was 10 mM ATP, 10 mM MnCl2, 10 mM MgCl2, 10 μM GGM-MU, 0.1% TX-100 in 0.1 M MES Buffer pH 6.5. Reactions were run at 37°C for 24 hr for HAP1 cells, 48 hr for fibroblasts, or 16 hr for skeletal muscle. Experiments were done in triplicate, with each replicate representing a separate plate of cells or animal. After POMK reaction, 6 μL 0.5 M EDTA was added to 30 μL of reaction supernatant, and the mixture boiled for 5 min. 25 μL of this mixture and added to 30 μL ddH20 in HPLC vial and run on an LC18 column of a reverse-phase HPLC (Shimadzu Scientific, Columbia, Maryland) with a 16% B med sensitivity gradient. The reaction was analyzed using a 4.6 × 250 mm Supelcosil LC-18 column (Supelco). Solvent A was 50 mM ammonium formate (pH 4.0), and solvent B was 80% acetonitrile in solvent A. Elution of the MU derivative was monitored by fluorescence detection (325 nm for excitation, and 380 nm for emission) and peak area used as a measure of activity. The enzymatic activity was calculated as the peak area of the product.

## B4GAT1 assay

For the assessment of endogenous B4GAT1 activity in skeletal muscle, Triton X-100-solubilized lysates (10 µl for human skeletal muscle or 40 µL for mouse skeletal muscle) were incubated in a volume of 50 µL (human skeletal muscle) for 12 hr at 37°C, with 0.4 mM Xylose-β-MU (Xyl-β-MU) and 10 mM Uridine diphosphate glucuronic acid (UDP-GlcA) in 0.1 M MES buffer, pH 6.0, at 5 mM $MnCl_2$, 5 mM $MgCl_2$, and 0.05% Triton X-100 (*Willer et al., 2014*). The reaction was terminated by adding 25 µL of 0.1 M EDTA and boiling for 5 min, and the supernatant was analyzed using an LC-18 column. Both the substrate Xyl-β-MU and the product GlcA-Xyl-β-MU were separated on a 16% acetonitrile isocratic gradient. Elution of the MU derivative product was monitored by fluorescence detection (325 nm for excitation, and 380 nm for emission). The percent conversion of substrate to product was used as the activity of the B4GAT1 in the 10 µL sample. The B4GAT1 activity then was normalized against the amount of protein measured in the 10 µL of sample using the DC protein assay (Bio-Rad, Hercules, CA) with BSA as the standard.

For assessment of B4GAT1 activity in HAP1 cells, the HAP1 WGA beads were incubated in a volume of 80 µL for 26 hr at 37°C, with 0.4 mM Xyl-β-MU and 10 mM UDP-GlcA in 0.1 M MES buffer, pH 6.0, at 5 mM $MnCl_2$, 5 mM $MgCl_2$, and 0.05% Triton X-100. The reaction was terminated by adding 25 µL of 0.1 M EDTA and boiling for 5 min, and the supernatant was analyzed using an LC-18 column. Elution of the MU derivative was monitored by fluorescence detection (325 nm for excitation, and 380 nm for emission) and peak area used as a measure of activity. The percent product was determined by taking the product peak area and dividing by the total peak areas of substrate plus product peak. Then this number was taken and multiplied by 100 for percent conversion to product.

## LARGE1 assay

For the assessment of endogenous LARGE1 GlcA-T activity in skeletal muscle, Triton X-100-solubilized lysates were incubated in a volume of 25 µL for 3 hr at 37°C, with 0.4 mM Xyl-α1,3-GlcA-β-MU and 10 mM UDP-GlcA in 0.1 M MES buffer, pH 6.0, at 5 mM $MnCl_2$, 5 mM $MgCl_2$, and 0.5% Triton X-100. The reaction was terminated by adding 25 µL of 0.1 M EDTA and boiling for 5 min, and the supernatant was analyzed using an LC-18 column. Elution of the MU derivative was monitored by fluorescence detection (325 nm for excitation, and 380 nm for emission) and peak area used as a measure of activity. The GlcA-T activity was assessed by subtracting the background observed in the negative control sample without donor sugar and normalized against the amount of protein measured using the DC protein assay (Bio-Rad).

For assessment of LARGE1 enzymatic activity in HAP1 cells, the Triton X-100-solubilized HAP1 cells were loaded onto WGA beads and processed as described for POMK assay above. The next day after wash, beads were incubated in a volume of 90 µL with 0.4 mM Xyl-α1,3-GlcA-β-MU and 10 mM UDP-GlcA in 0.1 M MOPS buffer, pH 6.0, at 5 mM $MnCl_2$, 5 mM $MgCl_2$, and 0.05% Triton X-100. The samples were run for 46 hr at 37°C. The reaction was terminated by adding 25 µL of 0.25 M EDTA and boiling for 5 min, and the supernatant was analyzed using an LC-18 column.

For the assessment of endogenous LARGE1 activity in fibroblasts, supernatants from Triton X-100-solubilized fibroblasts were (20 µL) directly used. Supernatants were incubated in a volume of 100 µL for 24 hr at 37°C, with 0.4 mM Xyl-α1,3-GlcA-β-MU and 10 mM UDP-GlcA in 0.1 M MES buffer, pH 6.0, at 5 mM MnCl2, 5 mM MgCl2, and 0.5% Triton X-100. The reaction was terminated by adding 25 µL of 0.1 M EDTA and boiling for 5 min, and the supernatant was analyzed using an LC-18 column.

Elution of the MU derivative was monitored by fluorescence detection (325 nm for excitation, and 380 nm for emission) and peak area used as a measure of activity. The percent product was determined by taking the product peak area and dividing by the total peak areas of substrate plus product peak. Then this number was taken and multiplied by 100 for percent conversion to product.

## B3GALNT2 assay

To assess B3GALNT2 activity in HAP1 cells, 20 µL of the WGA beads from HAP1 cells were incubated with a 20 µL volume of the reaction mix. The final volume of reaction buffer was 40 µL (30 µL reaction mixture and 10 µL WGA beads). The final concentrations were 10 mM $MgCl_2$, 10 mM $MnCl_2$, 0.1 M MES pH 6.5, 10 µM GM-MU, and 10 mM UDP-GalNAc. Reactions were run at 37°C for

72 hr. Experiments were done in triplicate, with each replicate representing a separate plate of cells. After B3GALNT2 reaction, 6 µL 0.5 M EDTA was added to 30 µL of reaction supernatant, and the mixture boiled for 5 min. 25 µL of this mixture and added to 30 µL ddH20 in HPLC vial and run on an LC18 column of a reverse-phase HPLC (Shimadzu Scientific) with a 16% B med sensitivity gradient. The reaction was analyzed using a 4.6 × 250 mm Supelcosil LC-18 column (Supelco, Bellefonte, PA). Solvent A was 50 mM ammonium formate (pH 4.0), and solvent B was 80% acetonitrile in solvent A. Elution of the MU derivative was monitored by fluorescence detection (325 nm for excitation, and 380 nm for emission) and peak area used as a measure of activity. The enzymatic activity was calculated as the peak area of the product.

## Digestion of α-DG with exoglycosidases

Exoglycosidase treatment was carried out as described previously (*Briggs et al., 2016*; *Salleh et al., 2006*; *Moracci et al., 2000*). *T. maritima* β-glucuronidase (*Salleh et al., 2006*; *Moracci et al., 2000*) (Bgus) and *S. solfataricus* α-xylosidase (Xylsa), both bearing a His-tag were overexpressed in *E. coli*, and purified using Talon metal affinity resin as described and activity determined as described (*Salleh et al., 2006*; *Moracci et al., 2000*) with some modifications. Briefly, the cell pellet was resuspended in 20 mM HEPES buffer (pH 7.3), 150 mM NaCl, 0.1% NP-40 and sonicated. After centrifugation (30 min at 40,000 x *g*), the crude extract was incubated with Benzonase (Novagen) for 1 hr at room temperature and then heat-fractionated for 10 min at 75°C. The supernatant was purified by using Talon metal affinity resin.

Samples to be digested by Bgus and Xylsa were exchanged into 150 mM sodium acetate (pH 5.5) solution and mixed with Bgus (0.45 U) and/or Xylsa (0.09 U), or no enzymes, and incubated overnight at 65°C. Samples were then run on SDS-PAGE, transferred to PVDF-FL (Millipore), and probed with anti-α-DG core antibody (AF6868) and anti-α-DG glycan antibody (IIH6). Enriched rabbit α-DG (100 µL of the 150 mM sodium acetate (pH 5.5) solution) was mixed with Bgus (0.45 U) and/or Xylsa (0.09 U), or no enzymes, and incubated overnight at 65°C. Samples were then run on SDS-PAGE, transferred to PVDF-FL (Millipore), and subjected to immunoblotting.

## Solid-phase assay

Solid-phase assays were performed as described previously (*Michele et al., 2002*; *Goddeeris et al., 2013*). Briefly, WGA eluates were diluted 1:50 in TBS and coated on polystyrene ELISA microplates (Costar 3590) overnight at 4°C. Plates were washed in LBB and blocked for 2 hr in 3% BSA/LBB at RT. The wells were washed with 1% BSA/LBB and incubated for 1 hr with L9393 (1:5000 dilution) in 3% BSA/LBB followed by incubation with Horseradish Peroxidase (HRP)-conjugated anti-rabbit IgG (Invitrogen, 1:5000 dilution) in 3% BSA/LBB for 30 min. Plates were developed with o-phenylenediamine dihydrochloride and $H_2O_2$, and reactions were stopped with 2 N $H_2SO_4$. Absorbance per well was read at 490 nm by a microplate reader.

## Statistics

The included Shimadzu post-run software was used to analyze POMK, LARGE1, and B4GAT1 activity in fibroblasts and mouse skeletal muscle, and the percent conversion to product was recorded. The means of three experimental replicates (biological replicates, where each replicate represents a different pair of tissue culture plates or animals, i.e. control and knockout) were calculated using Microsoft Excel, and the mean percent conversion to product for the WT or control sample (Control human fibroblasts or *Pomk*$^{LoxP/LoxP}$ skeletal muscle, respectively) reaction was set to 1. Percent conversion of each experimental reaction was subsequently normalized to that of the control, and statistics on normalized values were performed using GraphPad Prism 8. For analysis of POMK and LARGE1 activity in fibroblasts and mouse skeletal muscle, Student's t-test was used (two-sided). Differences were considered significant at a p-value less than 0.05. Graph images were also created using GraphPad Prism and the data in the present study are shown as the means + / - SD unless otherwise indicated. The number of sampled units, n, upon which we report statistics for in vivo data, is the single mouse (one mouse is n = 1).

For measure of POMK activity in HAP1 cells, the percent conversion from GGM-MU to GGM(P)-MU was first calculated using the included Shimadzu analysis software. The means plus standard deviations of the percent conversion to GGM(P)-MU for three experimental replicates was calculated

using GraphPad Prism 8. One-way ANOVA with the Dunnett's Method for Multiple Comparisons was performed, and the data for the *POMK* KO HAP1 sample set as the control. Differences were considered significant at a p-value less than 0.05. Graph images were created in GraphPad and show mean + / - SD.

To measure POMK activity in control and NH13-284 skeletal muscle, we only performed one experimental replicate due to the limited amount of sample available. To measure B4GAT1 activity, two technical replicates were performed from skeletal muscle. Protein concentration from control and NH13-284 skeletal muscle was also measured using two technical replicates. The percent conversion to product for the B4GAT1 reaction was divided by the protein concentration, and the values for these two technical replicates graphed using GraphPad Prism 8. The graph reported is shown as the mean + / - SD.

For flow cytometry analyses, six experimental replicates were performed, and the mean fluorescence intensity (MFI) reported. Statistics were performed using the Student's unpaired t-test, two-sided in GraphPad Prism eight and the values reported as mean + / - SD.

## NMR spectroscopy

1D $^1$H NMR spectra of the core M3 trisaccharides GGM-MU and GGMp-MU in the absence and presence of POMK or LARGE1 were acquired at 25°C on a Bruker Avance II 800 MHz NMR spectrometer equipped with a sensitive cryoprobe by using a 50 ms $T_2$ filter consisting of a train of spin-lock pulses to eliminate the broad resonances from the protein (**Mayer and Meyer, 2001**). *Danio rerio* POMK titrations were performed in 25 mM Tris (pH 8.0), 180 mM NaCl, and 10 mM MgCl$_2$ in 98% D$_2$O. LARGE1 titrations were performed in 20 mM HEPES, 150 mM NaCl, pH 7.3 in 90% H$_2$O/ 10% D$_2$O. The $^{13}$C and $^1$H resonances of the trisaccharides were reported previously (**Yoshida-Moriguchi et al., 2010**). The $^1$H chemical shifts are referenced to 2,2-dimethyl-2-silapentane-5-sulfonate. The NMR spectra were processed using NMRPipe (**Delaglio et al., 1995**) and analyzed using NMRView (**Johnson and Blevins, 1994**). The glycan binding affinity to POMK and LARGE1 was determined using glycan-observed NMR experiments as described previously (**Briggs et al., 2016**). For the resolved anomeric trisaccharide peak, the bound fraction was calculated by measuring the difference in the peak intensity in the absence (free form) and presence (bound form) of POMK or LARGE1, and then dividing by the peak intensity of the free form. To obtain dissociation constant, the data were fitted to the standard quadratic equation using GraphPad Prism (GraphPad Software). The standard deviation from data fitting is reported.

## Mass spectrometry

In order to generate DG fusion proteins for MS analyses, HAP1 cells were grown in IMDM with 10% FBS and 1% penicillin/streptomycin on p150 plates. When plates were 80% confluent, cells were washed twice with DPBS, media changed to serum-free IMDM with 1% penicillin/streptomycin (Invitrogen), and cells infected at high MOI (250–1000) of adenovirus expressing DG390TEVHis. Three days later, the medium was harvested and stored at 4°C until samples were ready for MS analysis.

Reductive elimination. Glycans were reductively eliminated from DG390 proteins and purified on a 50WS8 Dowex column, and the purified glycans were subjected to permethylation and purified according to published methods (**Jang-Lee et al., 2006**; **Zhang et al., 2014**). Briefly, the freeze-dried DG390 sample was dissolved in 55 mg/mL potassium borohydride in 1 mL of a 0.1 M potassium hydroxide solution. The mixture was incubated for 18 hr at 45°C and quenched by adding five to six drops of acetic acid. The sample was loaded on the Dowex column and subsequently eluted with 5% acetic acid. The collected solution was concentrated and lyophilized, and excessive borates were removed with 10% methanolic acetic acid.

Permethylation. For the permethylation reaction, three to five pellets per sample of sodium hydroxide were crushed in 3 mL dry dimethyl sulfoxide. Methyl Iodine (500 µL) as well as the resulting slurry (0.75 mL) were added to the sample. The mixture was agitated for 15 min and quenched by adding 2 mL ultrapure water with shaking. The glycans were extracted with chloroform (2 mL) and washed twice with ultrapure water. Chloroform was removed under a stream of nitrogen. The permethylated glycans were loaded on a C18 Sep-pak column, washed with 5 mL ultrapure water and successively eluted with 3 mL each of 15, 35, 50, and 75% aq. acetonitrile. The solutions were

collected and lyophilized. The lyophilized 35 and 50% fractions were dissolved in 50% aqueous solution of methanol and combined for MALDI analysis.

Mass spectrometry. A Bruker Autoflex III MALDI-TOF/TOF was used for acquisition of all MALDI MS data. An in-house made BSA digest was used to calibrate the MS mode. 3,4-diaminobenzophenone was used as the matrix. Permethylated samples were dissolved in 10 mL of methanol, and 1 μL of this solution was premixed with 1 μL matrix. 1 μL of this mixture was spotted on the plate.

## Acknowledgements

The BRC/MRC Centre for Neuromuscular Diseases Biobank is acknowledged for providing patients' serum and biopsy samples. The Muscular Dystrophy UK support to the GOSH Neuromuscular Centre is also gratefully acknowledged. We wish to thank Norma Sinclair, Patricia Yarolem, JoAnn Schwarting and Rongbin Guan for their technical expertise in generating transgenic mice.

## Additional information

### Funding

| Funder | Grant reference number | Author |
|---|---|---|
| Paul D. Wellstone Muscular Dystrophy Specialized Research Center grant | 1U54NS053672 | Kevin P Campbell |
| Great Ormond Street Hospital for Children | NHS Foundation Trust and University College | Silvia Torelli Francesco Muntoni |
| National Institute of General Medical Sciences | Medical Scientist Training Program Grant T32 GM007337 | Ameya S Walimbe |
| Howard Hughes Medical Institute | | Kevin P Campbell |

The funders had no role in study design, data collection and interpretation, or the decision to submit the work for publication.

### Author contributions

Ameya S Walimbe, Conceptualization, Formal analysis, Investigation, Methodology, Writing - original draft, Writing - review and editing; Hidehiko Okuma, Soumya Joseph, Jeffrey M Hord, Formal analysis, Investigation, Methodology, Writing - review and editing; Tiandi Yang, Takahiro Yonekawa, Investigation, Methodology, Writing - review and editing; David Venzke, Mary E Anderson, Silvia Torelli, Adnan Manzur, Saul Ocampo Landa, Junyu Xiao, Jack E Dixon, Investigation, Methodology; Megan Devereaux, Marco Cuellar, Sally Prouty, Methodology; Liping Yu, Conceptualization, Formal analysis, Investigation, Methodology, Writing - review and editing; Francesco Muntoni, Conceptualization, Supervision, Funding acquisition, Investigation, Methodology, Writing - review and editing; Kevin P Campbell, Conceptualization, Formal analysis, Supervision, Funding acquisition, Investigation, Methodology, Writing - original draft, Project administration, Writing - review and editing

### Author ORCIDs

Ameya S Walimbe https://orcid.org/0000-0002-3248-0761
David Venzke http://orcid.org/0000-0001-8180-9562
Junyu Xiao http://orcid.org/0000-0003-1822-1701
Jack E Dixon http://orcid.org/0000-0002-8266-5449
Kevin P Campbell https://orcid.org/0000-0003-2066-5889

### Ethics

Human subjects: All procedures performed in this study involving human participants were in accordance with the ethical standards of NHS Health Research Authority (REC ref: 06/Q0406/33). We acknowledge and thank the BRC/MRC Centre for Neuromuscular Diseases Biobank for providing

patients' serum and biopsy samples. We confirm that informed consent was provided to the patient and family regarding the nature of the genetic studies to be performed upon collection of samples and is available for our patient.

Animal experimentation: Animal experimentation: This study was performed in strict accordance with the recommendations in the Guide for the Care and Use of Laboratory Animals of the National Institutes of Health. All animal experiments were approved by the Institutional Animal Care and Use Committee (IACUC) protocols of the University of Iowa (#0081122).

### Decision letter and Author response
Decision letter https://doi.org/10.7554/eLife.61388.sa1
Author response https://doi.org/10.7554/eLife.61388.sa2

## Additional files
### Supplementary files
• Source data 1. Raw MALDI-TOF data of DG390 expressed in *FKTN* KO HAP1 cells (*Figure 6—figure supplement 2B*). Data were exported to the TXT format by FlexAnalysis 3.3 (Bruker Daltonics). The mass spectra in the article were zoomed into the range of m/z 750–950 to better present MS signals corresponding to core M3 glycan structures.

• Source data 2. Raw MALDI-TOF data of DG390 expressed in *POMK* KO HAP1 cells (*Figure 6—figure supplement 2A*). Data were exported to the TXT format by FlexAnalysis 3.3 (Bruker Daltonics). The mass spectra in the article were zoomed into the range of m/z 750–950 to better present MS signals corresponding to core M3 glycan structures.

• Transparent reporting form

### Data availability
All data generated or analysed during this study are included in the manuscript.

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

# Appendix 1

## Supplementary text

We transduced *POMK/DAG1* KO HAP1 and *POMK* KO HAP1 cells with an adenovirus expressing wild-type DG (Ad-DG). We observed a return of the laminin binding at 90-100 kDa in *POMK/DAG1* KO HAP1 cells (*Figure 7—figure supplement 4A*) and an increase in the corresponding IIH6 immunoreactivity and laminin binding in *POMK* KO HAP1 cells (*Figure 7—figure supplement 6A, B, C*), further indicating that the glycoprotein responsible is α-DG.

The binding of a Xyl-GlcA acid repeat of matriglycan to the LG4 domain of laminin α1 is calcium-dependent (*Hohenester, 2019*; *Briggs et al., 2016*; *Yoshida-Moriguchi and Campbell, 2015*). To test if the binding of the non-extended matriglycan is similarly calcium-dependent, we performed laminin overlays in the presence of 10 mM EDTA (*Figure 7—figure supplement 6D, E*). In both WT and *POMK* KO HAP1 cells, there was a complete absence of laminin binding in the presence of EDTA, indicating that laminin binding at 90-100 kDa is calcium-dependent and the glycan responsible is composed of Xyl-GlcA acid repeats.

Given the higher affinity of POMK for the unphosphorylated core M3 compared to the phosphorylated form (*Figure 8C*; *Figure 8—figure supplement 1A*), it is possible that POMK D204N, which is catalytically inactive, binds to GGM and increases the amount of core M3-modifed α-DG in the ER, thereby reducing the amount entering the Golgi. With a reduction in the amount of core M3-modified α-DG entering the Golgi, FKTN may be able to better modify GalNAc of the unphosphorylated core M3, thus enabling the formation of the matriglycan which enables laminin binding at 90-100 kDa in the patient's skeletal muscle. In *POMK* KO HAP1 cells alone, the non-extended matriglycan represents the amount formed when no POMK is present and transport of core M3-modified α-DG to the Golgi is not reduced. In support of this hypothesis, overexpression of POMK D204N in *POMK* KO HAP1 cells at a higher multiplicity of infection (MOI) of 10 leads to higher MW forms of matriglycan despite the catalytic inactivity of POMK D204N *in vitro* (*Figure 7—figure supplement 6F*). The higher MW of this form of matriglycan resembles that of the POMK D204N skeletal muscle. Alternatively, it is possible that POMK D204N remains attached to the unphosphorylated core M3 and this binary complex of POMK D204N and α-DG moves to the Golgi, where it can form a ternary complex with FKTN. The ternary complex composed of FKTN, POMK D204N, and α-DG enables FKTN to more efficiently elongate the core M3 leading to formation of the non-extended matriglycan. Further studies will be needed to determine the precise mechanism.

**Appendix 1—key resources table**

| Reagent type (species) or resource | Designation | Source or reference | Identifiers | Additional information |
|---|---|---|---|---|
| Genetic reagent *Mus musculus* | *Pomk*<sup>LoxP/LoxP</sup> ICR | This paper | Campbell Lab | Materials and Methods 'Generation of *Pomk*<sup>LoxP/LoxP</sup> Mice' |
| Genetic reagent *Mus musculus* | *Pax7*<sup>Cre</sup> C57BL/6J | The Jackson Laboratory, Bar Harbor ME. | JAX:010530, RRID:IMSRJAX:010530 | Pax7<sup>tm1(cre)Mrc</sup> |
| Genetic reagent *Mus musculus* | *Mck*<sup>Cre</sup> C57BL/6J | The Jackson Laboratory, Bar Harbor ME. | JAX:006475, RRID:IMSR_JAX: 006475 | B6.FVB(129S4)-Tg (Ckmm-cre)5Khn/J |
| Genetic reagent *Mus musculus* | *Large*<sup>myd</sup> C57BL/6J | The Jackson Laboratory, Bar Harbor ME. | JAX:000300, RRID:IMSR_JAX: 000300 | MYD/Le-Os +/+ Largemyd/J myd |
| Antibody | anti-DG (Sheep polyclonal) | R and D Systems | Cat# AF6868, RRID:AB_10891298 | WB (1:500) |
| Antibody | anti-α-DG (IIH6C4) (Mouse monoclonal) | DSHB Campbell Lab | Cat# IIH6 C4, RRID:AB_2617216 | WB (1:10-1:100) |

*Continued on next page*

*Appendix 1—key resources table continued*

| Reagent type (species) or resource | Designation | Source or reference | Identifiers | Additional information |
|---|---|---|---|---|
| Antibody | anti-myc clone 4A6 (Mouse monoclonal) | Millipore | Cat# 05–724, RRID:AB_309938 | WB (1:2000) |
| Antibody | anti-α-DG (IIH6C4) (Mouse monoclonal) | Millipore Campbell Lab | Cat# 05–593, RRID:AB_309828 | IF (1:1000-1:2000) |
| Antibody | anti-Laminin (Rabbit polyclonal) | Sigma-Aldrich | Cat# L9393, RRID:AB_477163 | WB (1:1000), Solid-Phase Assay (1:5000) |
| Antibody | anti-β-DG (Rabbit polyclonal) | Campbell Lab PMID:1741056 DOI:10.1038/355696a0 | AP83 | IF (1:50) |
| Antibody | anti-β-DG mouse IgM (Mouse monoclonal) | Leica Biosystems | Cat# NCL-b-DG, RRID:AB_442043 | IF (1:50 to 1:200) |
| Antibody | anti-Na$^+$,K$^+$ ATPase (Mouse monoclonal) | BD Biosciences | Cat# 610993 RRID:AB_398306 | WB (1:1000) |
| Antibody | anti-sheep IgG (Donkey polyclonal) | Rockland | Cat# 613-731-168, RRID:AB_220181 | WB (1:2000) |
| Antibody | anti-mouse IgG (H + L) (Donkey polyclonal) | LI-COR Biosciences | Cat# 926–32212, RRID:AB_621847 | WB (1:15,000), IF (1:800) |
| Antibody | anti-rabbit IgG (H + L) (Donkey polyclonal) | LI-COR Biosciences | Cat# 926–32213, RRID:AB_621848 | WB (1:15,000), IF (1:800) |
| Antibody | anti-mouse IgM (Goat polyclonal) | LI-COR Biosciences | Cat# 926–32280, RRID:AB_2814919 | WB (1:2500) |
| Antibody | anti-mouse IgG1 (Goat polyclonal) | LI-COR Biosciences | Cat# 926–32350, RRID:AB_2782997 | WB (1:2000, 1:10,000) |
| Antibody | anti-rabbit IgG (H+L) (Goat polyclonal) | Thermo Fisher Scientific | Cat# A-11034, RRID:AB_2576217 | IF (1:1000 to 1:2000) |
| Antibody | anti-mouse IgM (Goat polyclonal) | Thermo Fisher Scientific | Cat# A-21042, RRID:AB_2535711 | IF (1:1000 to 1:2000) |
| Antibody | anti-human FLJ23356 (Mouse monoclonal) | Novus | Cat# H00084197-M03, RRID:AB_2188284 | WB (1:500) |
| Commercial assay or kit | Creatine Kinase (CK) Liqui-UV Test | Fisher Scientific/ Stanbio | Cat# 22-022-630 | |
| Cell line (*Homo-sapiens*) | Parental cell line C631 | Horizon Discovery | Cat# C631 | Mycoplasma testing passed |
| Cell line (*Homo-sapiens*) | *POMK/DAG1* KO | Horizon Discovery | HZGHC001338c004, RRID:CVCL_TF19 | Authenticated by Sanger sequencing. Mycoplasma testing passed. |
| Cell line (*Homo-sapiens*) | *POMK/LARGE1* KO | Horizon Discovery | HZGHC007364c011 | Authenticated by Sanger sequencing. Mycoplasma testing passed. |
| Cell line (*Homo-sapiens*) | *POMK* KO | Horizon Discovery | HZGHC001338c004, RRID:CVCL_TF19 | Authenticated by Sanger sequencing. Mycoplasma testing passed. |
| Cell line (*Homo-sapiens*) | *POMK/ISPD* KO | Horizon Discovery | HZGHC001338c001, RRID:CVCL_TF18 | Authenticated by Sanger sequencing. Mycoplasma testing passed. |

*Continued on next page*

*Appendix 1—key resources table continued*

| Reagent type (species) or resource | Designation | Source or reference | Identifiers | Additional information |
|---|---|---|---|---|
| Cell line (*Homo-sapiens*) | *FKTN* KO | Horizon Discovery | HZGHC000721c010, RRID:CVCL_SN68 | Authenticated by Sanger sequencing. Mycoplasma testing passed. |
| Cell line (*Homo-sapiens*) | *LARGE1* KO | Horizon Discovery | HZGHC000122c007, RRID:CVCL_SV31 | Authenticated by Sanger sequencing. Mycoplasma testing passed. |
| Cell line (*Homo-sapiens*) | Primary dermal fibroblasts, human | ATCC | PCS-201–012 | |
| Cell line (*Homo-sapiens*) | Human fibroblasts (POMK D204N) | This paper | NH13-284 | Dubowitz Neuromuscular Center, Campbell Lab |
| Peptide, recombinant protein | β-Glucuronidase | PMID:16303119 DOI:10.1016/j.carres.2005.10.005 | | |
| Peptide, recombinant protein | α-Xylosidase | PMID:10801892 DOI:10.1074/jbc.M910392199 | | |
| Biological sample (*Homo-sapiens*) | Control human skeletal muscle | This paper | | Dubowitz Neuromuscular Center, Campbell Lab |
| Biological sample (*Homo-sapiens*) | Human skeletal muscle | This paper | (NH13-284, POMK D204N) | Dubowitz Neuromuscular Center, Campbell Lab |
| Chemical compound, drug | Purified *Danio rerio* POMK | PMID:27879205 DOI:10.7554/eLife.22238 | | |
| Chemical compound, drug | Purified mammalian dTMLARGE1 | PMID:22223806 DOI:10.1126/science.1214115 | | |
| Chemical compound, drug | GGM-MU and GGMp-MU | PMID:23929950 DOI:10.1126/science.1239951 | | |
| Chemical compound, drug | UDP-Xylose | CarboSource | https://www.ccrc.uga.edu/~carbosource/CSS_substrates.html | |
| Chemical compound, drug | 4-Methylumbelliferyl-β-D-xylopyranoside | Sigma/Millipore | Cat# M7008 | |
| Chemical compound, drug | UDP-Glucuronic acid | Sigma/Millipore | Cat# U6751 | |
| Chemical compound, drug | Uridine 5′-diphospho-N-acetylgalactosamine disodium salt | Sigma/Millipore | Cat# U5252 | |
| Chemical compound, drug | Uridine 5′-diphospho-N-acetylglucosamine sodium salt | Sigma/Millipore | Cat# U4375 | |
| Chemical compound, drug | 4-methylumbelliferyl α-(GlcNAc-β(1-4) Man) GM-MU | Sussex Research | https://www.sussex-research.com/ | |

*Continued on next page*

*Appendix 1—key resources table continued*

| Reagent type (species) or resource | Designation | Source or reference | Identifiers | Additional information |
|---|---|---|---|---|
| Chemical compound, drug | Xylose-α1,3-GlcA-β-MU | Sussex Research | https://www.sussex-research.com/ | |
| Chemical compound, drug | Pepstatin A | Sigma/Millipore | Cat# 516481 | |
| Chemical compound, drug | Calpain Inhibitor I (25 mg) | Sigma/Millipore | Cat# A6185 | |
| Chemical compound, drug | Aprotinin from bovine lung | Sigma/Millipore | Cat# A1153 | |
| Chemical compound, drug | Leupeptin (25 mg) | Sigma/Millipore | Cat# 108975 | |
| Chemical compound, drug | PMSF | Sigma/Millipore | Cat# P7626-25G | |
| Chemical compound, drug | Immobilon-FL PVDF | Sigma/Millipore | Cat# IPFL00010 | |
| Chemical compound, drug | Calpeptin | Fisher Scientific | Cat# 03-340-05125M | |
| Chemical compound, drug | Bis-acrylamide solution-30% (37:1) | Fisher Scientific/Hoefer | Cat# HBGR337500X | |
| Chemical compound, drug | Benzamidine Hydrochloride Hydrate | MP Biochemicals | Cat# 195068 | |
| Chemical compound, drug | WGA agarose bound | Vector Labs | Cat# AL-1023, RRID:AB_2336862 | |
| Chemical compound, drug | Precision Plus Protein All Blue Standards-500ul | Bio-Rad | Cat# 161–0373 | |
| Software, algorithm | SigmaPlot | SigmaPlot | RRID:SCR_003210 | |
| Software, algorithm | Excel | Microsoft | RRID:SCR_016137 | |
| Software, algorithm | GraphPad Prism | https://www.graphpad.com/scientific-software/prism/ | RRID:SCR_002798 | Version 8.3 |
| Software, algorithm | FlowJo | https://www.flowjo.com/solutions/flowjo/downloads | RRID:SCR_008520 | Version 7.6.5 |
| Software, algorithm | Li-Cor Image Studio Software | https://www.licor.com/bio/image-studio-lite/download | RRID:SCR_015795 | |
| Other | Streptavidin, Alexa Fluor 594 conjugate | Thermo Fisher Scientific | Cat# S11227 | IF (1:1000 to 1:2000) |
| Other | Adenovirus: DGC (DG, delta H30-A316) | PMID:21987822 DOI:10.1073/pnas.1114836108 | | |
| Other | Adenovirus: DG T317A | PMID:21987822 DOI:10.1073/pnas.1114836108 | | |
| Other | Adenovirus: DG T319A | PMID:21987822 DOI:10.1073/pnas.1114836108 | | |

*Appendix 1—key resources table continued*

| Reagent type (species) or resource | Designation | Source or reference | Identifiers | Additional information |
|---|---|---|---|---|
| Other | Adenovirus: DG T317A/319A | PMID:21987822 DOI:10.1073/pnas.1114836108 | | |
| Other | Adenovirus: DG Wild-Type (WT) | PMID:21987822 DOI:10.1073/pnas.1114836108 | | |
| Other | Adenovirus: POMK WT | PMID:27879205 DOI:10.7554/eLife.22238 | | |
| Other | Adenovirus: POMK D204A | PMID:27879205 DOI:10.7554/eLife.22238 | | |
| Other | Adenovirus: POMK D204N | This paper | Campbell Lab | Materials and methods 'Adenovirus Production' |
| Other | Adenovirus: DG390TEVHis | This paper | Campbell Lab | Materials and methods 'Adenovirus Production' |
| Other | Adenovirus: Fukutin | PMID:22522420 DOI:10.1038/ng.2252 | | |
| Other | Adenovirus: Isoprenoid Synthase Domain-Containing (ISPD) | PMID:22522420 DOI:10.1038/ng.2252 | | |
| Other | Adenovirus: LARGE1 | PMID:22522420 DOI:10.1038/ng.2252 | | |
| Other | NMR spectrometer | Bruker | Avance II 800 MHz | |
| Other | Rodent Treadmill | Columbus Instruments | Exer 3/6 Treadmill | |
| Other | Western Blot Imager | Li-Cor | Odyssey CLx | |
| Other | Mouse treadmill | Omnitech Electronics | Accupacer Treadmill | |
| Other | Isolated Mouse Muscle System | Aurora Scientific | 1200A | |
| Other | Mouse Grip Strength Meter | Columbus Instruments | 1027 Mouse | |
| Other | Prominence HPLC | Shimadzu | LC-20 system | |
| Other | Tabletop ultracentrifuge | Beckman Coulter | Optima max, 130K | |
| Other | Ultracentrifuge | Beckman Coulter | Optima-L-100 XP | |
| Other | Centrifuge | Beckman Coulter | Avanti J-E HPC | |
| Other | HPLC LC18 column | Supelco | 58368 | |

