## [Decision Letter]

**Acceptance summary:**

The study beautifully demonstrates that POMK is required for the synthesis of full- length and high-molecular weight forms of matriglycan, and that LARGE1 requires DGN to synthesize the short, non-extended form of matriglycan but needs both the DGN and the phosphorylated core M3 to generate full-length matriglycan.

**Decision letter after peer review:**

Thank you for submitting your article "POMK regulates dystroglycan function via LARGE-mediated elongation of matriglycan" for consideration by *eLife*. Your article has been reviewed by two peer reviewers, and the evaluation has been overseen by a Reviewing Editor and Michael Marletta as the Senior Editor. The following individuals involved in review of your submission have agreed to reveal their identity: Adnan Halim (Reviewer #1); James Paulson (Reviewer #2).

The reviewers have discussed the reviews with one another and the Reviewing Editor has drafted this decision to help you prepare a revised submission.

Our expectation is that authors will eventually carry out the additional experiments and report on how they affect the relevant conclusions either in a preprint on bioRxiv or medRxiv, or if appropriate, as a Research Advance in *eLife*, either of which would be linked to the original paper.

Both reviewers comment that the work is elegant and convincing in demonstrating the role of POMK in synthesis of matriglycan, the glycan polymer of α-dystroglycan (α-DG), and how its role in muscular dystrophy. Given that the number of comments/suggestions from reviewers is relatively few, we have opted to include them all in this decision letter, so that these can be addressed directly in revision, should you chose to revise the paper.

Summary:

Walimbe et al. aims to investigate the role of POMK in relation to matriglycan biosynthesis on core M3 *O*-Man glycans found on α-dystroglycan. Using cell lines, muscle specific *Pomk* KO mice, and patient muscle samples, the authors demonstrate convincingly that POMK driven mannose influences length of [GlcA-Xyl]_n_ repeats, as shown by Western blotting techniques. Furthermore, the authors characterize muscle physiology in mice with muscle specific KO of *Pomk*. Finally, binding affinities of Man/Man-6-P variants of core M3 to POMK and LARGE enzymes are determined, providing mechanistic details to support of their data. Collectively, this elegant study provides important insight into the role of POMK in matriglycan biosynthesis.

The synthesis the extended glycan polymer on α-DG, matriglycan, is controlled by over a dozen genes, and deficiencies in each lead to muscular dystrophies, illustrating the importance of the glycan as a scaffold in the muscle extracellular matrix, and the highly coordinated biosynthetic machinery involving each of these gene products needed to produce fully active matriglycan.

While the role of POMK was previously known to attach phosphate to the mannose of the trisaccharide core that bridges matriglycan to α-DG (-GalNAcβ1-3GlcNAcβ1-4Man-αDG), why the absence of this phosphate impacted the maturation of matriglycan was not known.

Through characterization of α-DG in a patient with POMK deficiency, and detailed biochemical and genetic experiments the authors show that the P-Man product is required for polymerization of matriglycan, even though the P-ManαThr/Ser product is remote from the action of the polymerase (LARGE) that adds the repeating disaccharide of matriglycan (-GlcAβ1-3Xylα1-3-)_n_. Here they clearly show that without POMK the matriglycan chains are short, resulting in α-DG of ~90 kDa, and with functional POMK, the extended matriglycan chains results in α-DG of ~150 kDa. A critical experiment is showing by NMR titration that LARGE binds to the tetrasaccharide core to initiate synthesis of the polymer, and that the phosphate product of POMK increases affinity of LARGE by an order of magnitude.

Overall the results are compelling, and the design of the experiments, and clear presentation of results and conclusions are very well done. The synthesis of matriglycan is arguably the most intricate post-translational modification in human biology, and the additional insights in the role of POMK represent a significant advance to the understanding of its role, and how its deficiency produces a mild form of muscular dystrophy.

Essential revisions:

Reviewers did not have specific required revisions.

Minor points:

1) The authors consistently observe a band at ~250 kDa in their laminin overlays (Fig 1D, 2B, 6C, 7A, 7D) but do not explain or comment as to why this reactivity is observed. It would be of general interest to know if this is unspecific binding, perhaps with a reference to a study or data to support this claim. Alternatively, if the authors consider the reactivity as specific, they could perhaps comment on the possibility that proteins other than α-dystroglycan could harbor laminin binding [GlcA-Xyl]_n_ repeats.

2) Annotation of genetic background of engineered cells is difficult to understand unless main text and figure legends are read in parallel. The authors should consider editing the annotation to make it clearer and uniform so that figures can be understood without necessarily having to read main text. For example, *Pomk^LoxP/LoxP^* (Fig 3B) speaks for itself, however, it is difficult to understand what e.g. *POMK/ISPD* KO and KO + Ad-ISPD (Fig 7C).

3) There appears to be an inconsistency for the size of α-dystroglycan/matriglycan reactivity in laminin overlays. For example, comparing Fig 5B and 5D, the majority of reactivity in 5B appears to be slightly below the 150 kDa marker - however, in 5D, the majority of reactivity is seen at-and-slightly-above 150 kDa. For HAP1 cells, reactivity is clearly below 150 kDa.

4) It is unclear why α-DG reactivity is observed above 150 kDa in Fig 5A when IIH6 detects α-aDG at <150 kDa.

5) Fig 6D-E, mass spectrometry: The authors should include Y-axis to allow the reader to see relative abundances of annotated *O*-glycans. The authors should also annotate all peaks (at-and-above the intensity of m/z 873.5) with their m/z values, including the abundant peaks that have been cropped. MALDI-TOF does not allow the authors to determine anomeric or epimeric configuration of annotated *O*-glycans (color coded), the authors should therefore consider including a sentence in figure legend to clarify this limitation to the reader.

6) Subsection “POMK assay”: It is unclear to me why the authors perform WGA enrichment of POMK from HAP1 and fibroblast cells, but not from mouse and human skeletal muscle.

7) Subsection “Mass Spectrometry” – Permethylation: Methyl Iodine (500 mL) seems excessive?

8) Subsection “Mass Spectrometry” – –Permethylation: Permethylated glycans where purified on C18 Sep-pak columns and eluted in different fractions, however, it is unclear which fractions where analyzed by MALDI-TOF.

9) Subsection “Data Availability”: Raw mass spectrometry data is not made available for readership. Will authors deposit their MALDI-TOF data in a public repository?

10) There should be a better discussion of the binding of LARGE to DGN and the P-Man core. For example, how far is the DGN binding site from the glycan chains in the middle of α-DG. Do the authors envision that LARGE forms a complex with simultaneous binding to both DGN and a and P-Man, or that DGN binding recruits LARGE, and it dissociates to bind P-Man and while bound to extend matriglycan? At a minimum, it would be helpful to the reader to better describe the physical relationships between DGN and the sites of attachment of the glycan cores to α-DG.

---

## [Author Response]

Since there were only minor comments/suggestions for the author to address in revision there is no accompanying author response.